



# Carbon and nitrogen turnover in the Arctic deep sea: in situ benthic community response to diatom and coccolithophorid phytodetritus

Ulrike Braeckman[1,2], Felix Janssen[3], Gaute Lavik[2,], Marcus Elvert[4], Hannah Marchant[2], Caroline Buckner[2], Christina Bienhold[2,3], Frank Wenzhöfer[2,3]

[1]Marine Biology Research Group, Ghent University, Krijgslaan 281 S8, 9000 Gent, Belgium
[2]Max Planck Institute for Marine Microbiology, Celsiusstrasse 1, 28359 Bremen, Germany
[3]HGF-MPG Group for Deep Sea Ecology and Technology, Alfred Wegener Institute Helmholtz Centre for Polar and Marine Research, Am Handelshafen 12, 27570 Bremerhaven, Germany
[4]MARUM Center for Marine Environmental Sciences and Department of Geosciences, University of Bremen, Leobener Strasse 8, 28359 Bremen, Germany

*Correspondence to*: Ulrike Braeckman (ulrike.braeckman@ugent.be)

**Abstract.** In the Arctic Ocean, increased sea surface temperature and sea ice retreat have triggered shifts in phytoplankton communities. In Fram Strait, coccolithophorids have been occasionally observed to replace diatoms as the dominating taxon of spring blooms. Deep-sea benthic communities depend strongly on such blooms but with a change in quality and quantity of primarily produced organic matter [OM] input, this may likely have implications for deep-sea life. We compared the in situ responses of Arctic deep-sea benthos to input of phytodetritus from a diatom (*Thalassiosira* sp.) and a coccolithophorid (*Emiliania huxleyi*) species. We traced the fate of $^{13}$C and $^{15}$N labelled phytodetritus into respiration, assimilation by bacteria and infauna in a 4 d and 14 d experiment. Bacteria were key assimilators in the *Thalassiosira* OM degradation whereas Foraminifera and other infauna were at least as important as bacteria in the *Emiliania* OM assimilation. After 14 d, 5 times less carbon and 3.8 times less nitrogen of the *Emiliania* detritus was recycled compared to *Thalassiosira* detritus. This implies that the utilization of *Emiliania* OM may be less efficient than for *Thalassiosira* OM. Our results indicate that a shift from diatom-dominated input to a coccolithophorid-dominated pulse could entail a delay in OM cycling, which may affect bentho-pelagic coupling.



## 1 Introduction

The Arctic is warming more rapidly than the global average (IPCC, 2014) with an increase in air temperature of roughly 2°C since 1900. In the Arctic Ocean, this has led to a concomitant rise in sea surface temperature of 1.5°C, most distinctly since the 1980s (Polyakov et al., 2013). As a result, summer sea-ice extent is presently decreasing at a rate of more than 10 % per decade (Comiso, 2010).

Fram Strait, located in the transition zone between the northern North Atlantic and the central Arctic Ocean, is the sole deep gateway where these two hydrographic regimes partly converge. In eastern Fram Strait, the West Spitsbergen Current (WSC) transports relatively warm (2.7-8°C) Atlantic water into the Arctic Ocean, while in the western strait, the East Greenland Current (EGC) carries colder (-1.7-0°C) polar water in the upper 150 m towards the south (Nöthig et al., 2015). During the last decade, the mean temperature of Atlantic water entering

the Arctic Ocean with the WSC increased by more than 0.05°C yr$^{-1}$ (Beszczynska-Möller et al., 2012).

Phytoplankton size and community structure are directly influenced by temperature, and smaller temperate species may become established within the phytoplankton community in areas with increased temperatures and less sea ice (Hilligsøe et al., 2011; Morán et al., 2010). This phenomenon also occurred in the eastern Fram Strait, where previously, phytoplankton communities were typically dominated by diatoms. However, during recent warmer

years, diatom blooms became more mixed with *Phaeocystis pouchetti* (Nöthig et al., 2015; Soltwedel et al., 2015). Also *Emiliania huxleyi* (Prymnesiophyceae)–dominated coccolithophorid blooms have been observed between 2000-2005 – especially in 2004-, which has been attributed to northward transport of the species into Fram Strait by means of the North Atlantic Current and WSC. This 'Atlantification' with a combined change in water temperature and water mass origin has been suggested as one possible scenario for a community shift in

phytoplankton communities in Fram Strait (Bauerfeind et al., 2009).

As the recycling of the phytoplankton bloom mainly occurs in surface waters, on average only a small fraction of the primary produced organic carbon (< 5 %) is exported to the deep-sea (Gooday and Turley, 1990; Schlüter et al., 2000). Earlier observations in Fram strait have shown that the Particulate Organic Carbon (POC) export flux at 300 m is very similar to the diffusive oxygen uptake in sediments (Bauerfeind et al., 2009), which suggests that,

once passed through the photic zone of the surface waters, degradation in the aphotic zone of the pelagic is negligible (Cathalot et al., 2015). The deposition of phytodetritus from surface water primary production is of crucial importance for the deep-sea benthos (Graf, 1989) that profoundly depends both on the quality and quantity of the settling food (Billett et al., 2010; Ruhl and Smith, 2004; Smith et al., 2013). It is expected that a change in quality and quantity of primarily produced organic matter [OM] input may have implications for deep-sea life.

A powerful approach to quantify the processing of a food source in benthic food webs is to label the organic matter (OM) with stable carbon and/or nitrogen isotopes. This technique allows for tracing the fate of the food source into different carbon and/or nitrogen pools representative of different biological processes and groups of organisms. This approach has been successfully applied in a wide range of study areas, from temperate, estuarine ecosystems to the abyssal sea floor (Woulds et al., 2009, 2016). However, most studies target assimilation by

specific size classes of organisms, e.g. macrofauna, or specific taxonomic groups such as Foraminifera (Enge et al., 2011; Nomaki et al., 2005) or Nematoda (Guilini et al., 2010; Ingels et al., 2010) and only few studies consider other mineralization pathways (e.g. respiration) (Buhring et al., 2006; Evrard et al., 2012; Woulds et al., 2016).





Experimental food web studies in the Arctic deep-sea with labelled food sources are scarce and have focussed on uptake of DOC, bacteria or diatom detritus by nematodes (Guilini et al., 2010; Ingels et al., 2010) or phytoplankton

and ice algae by macrofauna (Mäkelä et al., 2017). The latter studies are ship-based experiments, an approach which can involve decompression of the samples and is therefore prone to introducing biases, e.g. by changing the activity of the benthic biota. This type of studies shows that infauna selects its food sources specifically or is indifferent: macrofauna seems to have a preference for ice algae over phytoplankton in shallower Arctic water (McMahon et al., 2006; Sun et al., 2007) but displays a dietary plasticity in Arctic deep-sea sediments, assimilating

both ice algae and phytoplankton efficiently (Mäkelä et al., 2017). Also in the abyssal Pacific, macrofauna did not show preference for any type of food source when phytodetritus was readily available, whereas foraminifera selected coccolithophore nitrogen over diatom nitrogen (Jeffreys et al., 2013).

In this study, we compare the *in situ* mineralization pathways of phytodetritus of a traditionally prevailing primary producer with a food source that will possibly dominate in the near future as a consequence of global change

effects on phytoplankton communities. Such a shift may entail significant effects on the ecosystem since the availability and degradation patterns of different food sources might differ (Mäkelä et al., 2017; Ruhl and Smith, 2004; Smith et al., 2013). The previously dominant diatom *Thalassiosira*, for example, is protected by a silica wall which can be easily dissolved by bacteria (Bidle and Azam, 1999). The coccolithophorid *Emiliania huxleyi* on the other hand, consists for a substantial part of inorganic carbon (coccoliths) which can form a physical barrier to

bacterial degradation (Engel et al., 2009) and produces DMSP-related substances that act as anti-grazing compounds for zooplankton (Hansen et al., 1996), both characteristics that could imply a delay in its decomposition. Therefore, we hypothesize that a shift in phytoplankton communities from diatoms to coccolithophorids could have implications for the deep-sea benthos OM assimilation and may delay or impede the mineralization processes at the Arctic deep-sea floor.

To test this, we conducted a comparative *in situ* experiment at the Arctic deep-sea floor at 2500 m water depth, where we provided $^{13}$C and $^{15}$N labelled phytodetritus of either the diatom *Thalassiosira* sp. or the coccolithophorid *Emiliania huxleyi* to the benthic community. We tracked the degradation and processing pathways during a short (4d) and long term (14d) experiment in bacteria, infauna (>250 µm), and dissolved pools of inorganic carbon and nitrogen in the pore water and the overlying water. These experiments allow us to evaluate the general response

of Arctic deep-sea communities to food pulses as well as specific effects of changes in phytodetritus composition.

## 2 Materials and Methods

### 2.1 Study site

This study was conducted during RV *Maria S. Merian* expedition MSM29 in June-July 2013 at station S2 (MSM29/423-4 and MSM29/435-1), 78.776° N 5.2571° E, 2500 m water depth) of the long term deep-sea

observatory HAUSGARTEN in Fram Strait. The deep waters of the area are characterized by a constant temperature of -0.7°C and salinity of 35 (Beszczynska-Möller et al., 2012). The annual spring phytoplankton bloom in 2013 occurred in June (NASA Ocean Color, https://oceancolor.gsfc.nasa.gov/) but an earlier bloom in April had arrived at the sea floor with 9 mg POC m$^{-2}$ d$^{-1}$ one month before the start of the experiment (Salter et al. unpublished results). This implies that the benthic communities had been able to graze on some fresh OM prior to



our experiment, but not yet on the main annual food pulse. The observed response rates are therefore assumed to be representative rates of the processing of freshly deposited phytodetritus.

Oxygen penetrates deeper than 60 mm (Cathalot et al., 2015; Hoffmann et al., 2018) in the muddy sediment. Biomass in these sediments is dominated by bacteria (93 %), followed by macrofauna (6.2 %), megafauna (4 %) and nematode-dominated meiofauna (0.7 %) (van Oevelen et al., 2011).

**2.2 Culturing of algae**

The diatom *Thalassiosira* sp. and the coccolithophorid *Emiliania huxleyi* were cultured at 15°C (continuous light) in artificial seawater amended with f/2 medium (Guillard, 1975), $^{13}$C-bicarbonate, and $^{15}$N-nitrate (99 atom % $^{13}$C-enriched NaHCO$_3$ and Na$^{15}$NO$_3$; Cambridge Isotope Laboratories). The algal material was harvested by centrifugation (3500-5500 × g, 10-20 min). The harvest was frozen at -26°C. *Thalassiosira* contained 33 atom%

$^{13}$C and 35 atom% $^{15}$N, while *Emiliania* contained 31 atom% $^{13}$C and 43 atom% $^{15}$N (measured via Elemental Analyser – Isotope Ratio Mass Spectrometry – EA-IRMS, see below). Their corresponding molar C:N ratio was 13.3 (*Thalassiosira*) and 12.5 (*Emiliania*). Total carbon (TC) and organic carbon content (TOC) of the cultures were measured on subsamples of parallel cultures that were unmodified (TC) or acidified (TOC), respectively, prior to measurement using EA-IRMS. The corresponding TOC of the algae was 78 % (*Emiliania*) and 95 %

(*Thalassiosira*). Before use, thawed algae were washed 3 times in 0.2 µm filtered seawater to remove labelled DIC and dissolved inorganic nitrogen (DIN) that may have stuck to the exterior of the organisms. These washing steps also entailed a loss of DOM and total dissolved nitrogen (TDN). DOC and TDN were measured in the supernatant from the 3 washes with a TOC-VCPH Shimadzu instrument (Stubbins and Dittmar, 2012) with a precision better than 3%. In terms of carbon, on average 10.4 % of the harvested algae was lost as DOC, whereas in terms of

nitrogen, on average 53.8 % of the harvested algae was lost as TDN. The amount of phytodetritus reported below is corrected for this DOM loss. Although the algae were not cultured in axenic conditions, fatty acid (FA) analysis of the algal material did not show presence nor labelling of bacteria-specific iso- and anteiso-branched phospholipid-derived FAs (PLFAs i14:0, i and ai15:0, i16:0), which suggests that the bacterial contamination of the algae cultures was negligible.

**2.3 Experimental set-up**

In June 2013, two landers equipped with three benthic chambers (20 cm x 20 cm) were deployed at the deep-sea floor. One lander deployment lasted for 4 days, the other one for 14 days to investigate the temporal evolution of the OM processing. After arrival of the lander systems at the seafloor, a motorized drive lowered the chambers approx. 10 cm into the sediment to enclose 400 cm$^2$ of sediment together with approx. 20 cm of overlying water.

Labelled algae suspensions were added to two of the chambers of each experiment by means of an automated dispenser (Witte et al., 2003c). 69 mmol C$_{org}$ m$^{-2}$ (2.9 mmol N m$^{-2}$) of *Thalassiosira* sp. was added to one of the chambers, while 37 mmol C$_{org}$ m$^{-2}$ (2 mmol N m$^{-2}$) of *Emiliania huxleyi* was added to a second chamber. The third chamber was left unamended (control). These additions correspond to 0.85 g C$_{org}$ m$^{-2}$ (0.90 g C m$^{-2}$; 0.04 g N m$^{-2}$) for *Thalassiosira* and 0.45 g C$_{org}$ m$^{-2}$ (0.59 g C m$^{-2}$; 0.03 g N m$^{-2}$) for *Emiliania*. The treatments are further referred

to as '*Thalassiosira*', '*Emiliania*' and 'control'. The amount of added carbon was aimed to simulate the annual carbon deposition in the area. However, logistical issues with cultivation of *Emiliania* unfortunately compromised the addition of a C dose comparable to *Thalassiosira*. The additions correspond to 29-43 % (*Thalassiosira*) and





15-23 % (*Emiliania*) of the average total annual carbon deposition in the area (around 2-3 g $C_{org}$ m$^{-2}$) (Bauerfeind et al., 2009; Soltwedel et al., 2015) and represent 4.5 (*Emiliania*) to 8 (*Thalassiosira*) times the amount of POC

that had already arrived during the first sedimentation peak in late May 2013.

To verify the release of algae at the sea floor, small Teflon-coated laboratory stirring bars had been added to each algae suspension, which were recovered together with the incubated sediments after the deployment. During the course of the incubation, the overlying water was continuously stirred, to mimic the hydrodynamic conditions. An $O_2$ optode (type 4330, Aanderaa, Bergen, Norway) attached to the upper lid of the chamber monitored oxygen

concentration and temperature in the overlying water. An automated syringe sampler took 50 mL subsamples of the overlying water at seven regularly spaced time points by means of glass syringes. At the end of the incubation, the incubated sediments were enclosed from below with a motorized lid before the chambers were retracted from the seafloor. Upon acoustic release from the ship, the lander including the three chambers with the enclosed sediment and the overlying water was recovered. On board, the height of the overlying water body was measured

with a ruler at ~8 positions within each chamber. This allowed the estimation of the volume of overlying water, necessary to calculate the fluxes at the sediment-water interface.

Due to logistic and technical constraints associated with in situ experiments in the deep-sea, our experiments are unreplicated. We have chosen to invest in observing temporal dynamics (short vs. long term incubations) which would allow to unravel time-dependent aspects of the mineralization pathways (carbon respiration and nitrogen

mineralisation for different food web compartments) and link these with degradation processes. This also gave us the chance to look into the effect of different food sources, which is highly relevant in light of the expected changes in primary producer communities. We acknowledge that spatial variability may have contributed to differences observed in the algae and time treatments. However, we assumed that the reliability of our observations on the effect of the algae species were supported when differences between algae treatments observed in the short

incubation were confirmed by the long incubation. A similar approach was applied in earlier comparable deep-sea (lander) experiments which provided valuable new insights (Guilini et al., 2010; Woulds et al., 2007).

### 2.4 Sampling

Upon lander retrieval, the syringe samples of the overlying water and the sediments were immediately processed. Subsamples of the water samples in the syringes were transferred to 12 mL exetainers and fixed with 100 μL

HgCl$_2$ for later analysis of $^{13}$C-DIC and $^{15}$N-DIN. Another 1.5 mL subsample was transferred to Zinsser vials and fixed with 25 μL HgCl$_2$ for total DIC analysis. The samples were stored at 4°C until analysis. The remaining volume was stored frozen for nutrient analysis (without filtration; however, suspended matter in these overlying waters is negligible).

The sediments were subsampled at a vertical resolution of 0-1 cm, 1-2 cm, 2-3 cm and 3-5 cm and each horizon

was carefully mixed prior to subsampling. For pore water analysis of $^{13}$C-DIC and $^{15}$N-DIN, 5mL of the sediment from each horizon was added to 12 mL exetainers pre-filled with He-purged water. These samples were fixed with 100 μL HgCl$_2$. For total DIC and DIN analysis, pore water was extracted by centrifugation and stored as described for the overlying water samples.



The remainder of the sediment was subsampled for pigments (1 mL, frozen at -80°C), TOC and Total Nitrogen
(TN) (2 mL, frozen at -20°C), bacterial counts (Acridine Orange Direct Counts; AODC) (1 mL, stored in 9 mL 2
% formaldehyde-seawater solution), bacterial enzymatic activity (analysed directly on board), PLFAs (20 mL in
glass bottles, stored at -20°C) and fauna analysis (remaining volume, about 60 mL; stored in 4 % borax buffered
formaldehyde solution).

Due to tilting of the 4 d incubation lander by the *in situ* microprofiler, the *Emiliania* chamber contained less
sediment and only the 0-1 cm sediment horizon could be sampled. As will be discussed further, this undersampling
of the sediment in the 4 d *Emiliania* chamber results in an underestimation of the labelled phytodetritus of < 10 %.

### 2.5 $^{13}$C-DIC analyses

A 1.5 mL water subsample was transferred to a 12 mL exetainer and degassed with helium. 150 μL of 20 % $H_3PO_4$
was added and the samples were left overnight so that the DIC in solution would fill the headspace in the form of
$CO_2$. This headspace was then measured 8 times and the stable isotope ratio ($^{13}$C/$^{12}$C) was determined via a Gas
Bench II (Thermo Electron, Bremen Germany) coupled to an IRMS (Thermo Quest Delta Plus, Thermo Electron,
Bremen Germany). $CO_2$ was used as a reference gas and bicarbonate standards with concentrations similar to that
of the samples were also added as reference. The standard deviation of the measurements was < 0.001 at%. To
determine carbon respiration, the change in the stable isotope ratio (excess atomic % $^{13}$C) from the initial time
point was multiplied by the total DIC concentration measured by flow injection analysis with conductivity
detection (Hall and Aller, 1992) with a precision better than 2%.

### 2.6 $^{15}$N-DIN analyses

Subsamples (overlying water: 4 mL, pore water: 2 mL diluted with 2 mL milli-Q) were transferred to exetainers
and degassed with helium. $^{15}NH_4^+$ was oxidised with hypobromite to $N_2$ (Preisler et al., 2007; Warembourg, 1993).
A second set of 4.5 mL subsamples was also transferred to exetainers and $^{15}NO_x^-$ ($^{15}NO_2^-$ + $^{15}NO_3^-$) concentrations
were determined after conversion to $N_2$. $^{15}NO_3^-$ was reduced to $^{15}NO_2^-$ using spongy cadmium, followed by
$^{15}NO_2^-$ conversion to $N_2$ using sulfamic acid (Füssel et al., 2012). The stable isotope ratios of $^{28}N_2$, $^{29}N_2$ and $^{30}N_2$
were analyzed by a GC-IRMS (VG Optima, Micromass, Manchester, UK). The standard deviation of the
measurements was < 0.001 at%. Concentrations and rates of $^{29}N_2$ and $^{30}N_2$ production were calculated from the
excess relative to air, as explained in detail in Holtappels et al. (2011), and the efficiency of $^{15}NO_x^-$ or $^{15}NH_4^+$
conversion to $N_2$ was verified using known concentrations of $^{15}NH_4^+$ or $^{15}NO_3^-$.

### 2.7 General sediment analyses

Sediment median grain size was measured by Laser diffraction using a Malvern Mastersizer 2000G, hydro version
5.40. Pigments were extracted with 90% acetone and measured with a TURNER fluorimeter (Holm-Hansen et al.,
1965; Yentsch and Menzel, 1963). TOC and TN were measured with an Elemental Analyser after acidification of
the sediments to remove inorganic carbon in aliquots of approx. ~15 mg (dry weight). Another sediment subsample
was taken for stable carbon and nitrogen isotope analysis (see below).




### 2.8 Acridine Orange Direct Cell Counts and bacterial enzymatic activity

Microbial (bacterial and archaeal) cell numbers were determined using AODC. 1 ml or 2 ml sediment were fixed
with sterile filtered formalin/seawater at a final concentration of 2% and stored at 4°C. Samples were processed as
previously described (Hoffmann et al., 2017), and two replicate filters were counted for each sample using an
epifluorescence microscope (Axiophot, Zeiss). Extracellular enzymatic turnover rates in the sediment were
determined on board using the fluorogenic substrate fluorescein-di-acetate (FDA) as an indicator of the potential
hydrolytic activity of bacteria (Köster et al., 1991) but only from the short term (4 d) experiment.

### 2.9 DNA extraction, PCR, amplicon sequencing, and sequence processing

DNA was extracted from 0.5 g sediment using the MoBio PowerSoil Kit (MO BIO Laboratories, Inc.) following
manufacturer's instructions and eluted in a final volume of 60 μl TE-buffer (10 mM Tris-Cl, pH 8.0, 1 mM EDTA),
instead of solution S5 provided in the kit. DNA was quantified using a microplate spectrometer (InfiniteR 200
PRO NanoQuant, TECAN Ltd, Switzerland). Amplicon libraries of the bacterial V4-V6 region of the 16S rRNA
gene were generated according to the protocol recommended by Illumina (16S Metagenomic Sequencing Library
Preparation, Part #15044223, Rev. B), using the primers S-D-Bact-0564-a-S-15 and S-*Univ-1100-a-A-15 primer
(Klindworth et al., 2013). Sequencing was performed on an Illumina MiSeq 161 platform in 2x300 cycles paired
end runs. Raw paired-end sequences have been submitted to ENA under INSDC accession number
PRJEB25160(under embargo, will be released upon acceptance using the data brokerage service of the German
Federation for Biological Data (GFBio (Diepenbroek et al., 2014)).
Sequence processing included the following steps: Primer sequences were removed using *cutadapt* (v. 1.8.1
(Martin, 2011)). Forward and reverse reads were merged using *pear* (v. 0.9.5 (Zhang et al., 2013)); all sequences
were trimmed and quality filtered using *trimmomatic* (v. 0.32 (Bolger et al., 2014)). Reads were then clustered
into OTUs by applying a local clustering threshold of d=1 and the fastidious option in *swarm* (v. 2.1.1 (Mahé et
al., 2015)). The SINA aligner (v.1.2.10 (Pruesse et al., 2012)) was used to align and classify the seed sequence of
each OTU with the SILVA SSU database release 123 (Quast et al., 2012). We removed all OTUs that were
classified as chloroplasts, mitochondria, archaea, or those that could not be classified at domain level from further
analysis. Absolute singletons, *i.e.* sequences occurring only once in the entire dataset, were removed from further
analyses. This resulted in a final number of 3635 bacterial OTU.

### 2.10 Sediment lipid extraction and FA analyses

The bacterial incorporation of added phytodetritus was estimated through the isotope enrichment of bacterial-
specific PLFAs (Boschker and Middelburg, 2002). Two main biomarkers were chosen for the analysis: iC15:0 and
ai15:0 because of their specificity for bacteria and presence in all samples. Lipid extraction was performed using
a modified method from Bligh & Dyer (Bligh and Dyer, 1959) according to Sturt et al. (2004). In short, lipids were
extracted using a mixture of methanol, dichloromethane and phosphate buffer to pH 7.4 or trichloroacetic acid
(2:1:0.8 v/v). From this total lipid extract, an aliquot (1/2) was saponified using 6% KOH in methanol, after which
neutral lipids were released with hexane and subsequently removed (Elvert et al., 2003). The remaining methanolic
water phase was acidified to pH 1 and free FAs were extracted with hexane. FAMEs were identified via GC-MS
(Thermo Quest Trace GC with Trace MS) and concentrations determined by gas chromatography (GC)–flame
ionization detection (Thermo Finnigan Trace GC) relative to the internal standard (IS) 2Me-octadecanoic acid




added prior to extraction. Corresponding stable carbon isotope compositions of FAMEs were determined by GC-IRMS (Thermo Scientific V Delta Plus with Trace GC ultra, connected via GC Isolink and ConFlo IV interfaces) using $CO_2$ as a reference, and cross-checked against the known $\delta^{13}C$ value of the IS FA. $\delta^{13}C$ values have been corrected for the methyl group added during derivatization. $\delta^{13}C$ values have an analytical error of 1 per mil based on duplicate injection of selected samples.


### 2.11 Fauna analysis

Following staining with rose Bengal, sediments were sieved on a 250 µm mesh to retrieve macrofauna and the larger fraction of the meiofauna. We did not consider the meiofauna fraction < 250 µm since earlier studies in the research area have proven that nematodes constitute the bulk of this fraction and their share in $^{13}C$ assimilation is negligible (Guilini et al., 2010; Ingels et al., 2010). Organisms were sorted on higher taxon level (Nematoda, Foraminifera, Polychaeta, Bivalvia, Amphipoda, Tanaidacea, Porifera) and in most cases pooled over several depth layers to reach sufficient biomass. After drying at 60 °C of each taxon fraction, samples were prepared for stable carbon and nitrogen isotope analysis via EA-IRMS (see below). Faunal biomass was determined via C content values from the IRMS, combined with faunal abundance.



### 2.12 EA-IRMS analysis

Oven-dried sediment and fauna samples were decalcified overnight with the fumes of 37 % HCl in a desiccator. Freeze-dried algae samples were used without decalcification. Prepared samples were packed into tin cups and analyzed by a Thermo Flash EA 1112 elemental analyzer coupled to an isotopic ratio mass spectrometer (Thermo Delta Plus XP, Thermo Fisher Scientific, Waltham, MA, USA). Caffeine was used as a standard for isotope correction and C/N quantification of bulk carbon and nitrogen assimilation. Precisions of the caffeine measurements were: C = 1.07 ± 4.57 $10^{-5}$ at% and N = 0.37 ± 6.22 $10^{-5}$ at% (n = 23).


### 2.13 Calculations

Rates of total oxygen uptake (TOU), $^{13}C$-DIC and $^{15}N$–$NO_x^-$ accumulation in the overlying water were calculated from the slope of linear regressions of concentration as a function of time. Only significant (p < 0.05) and linear accumulation or consumption (checked by linear regression) of the mentioned species was considered.


$^{15}N$-$NH_4^+$ concentrations in the water column did not follow a linear increase or decrease over time, but rather an initial increase, followed by a decrease. Therefore, the minimum turnover of $^{15}N$-$NH_4^+$ was calculated as the sum of the initial accumulation and the following consumption in the water column.

The carbon accumulation in the DIC pool in the pore water, bulk sediment and fauna was calculated as the product of the excess atom% $^{13}C$ and the carbon content of the sample, divided by the atom% $^{13}C$ of the labeled algae:


$$C - accumulation \ (\mu g\ C) = \frac{(\text{atom\% }^{13}C_{sample} - \text{atom\% }^{13}C_{background})}{\text{atom\% algae} \times \text{TOC}_{sample}} \quad \text{(Eq. 1)}$$

Calculations for $^{15}N$ were made accordingly.





$^{13}$C label incorporation into bacterial biomass was based on the bacterial PLFAs that were present in all chambers and depth horizons (i15:0, ai15:0) (Boschker and Middelburg, 2002). For each bacterial PLFA, $^{13}$C label
incorporation was calculated as:

$$I_{PLFA} = E_{PLFA} \times PLFA_{carbon\ concentration} \qquad (Eq.\ 2)$$

Where Excess $^{13}$C (E) is given by the difference in fraction $^{13}$C in the sample ($F_{sample}$) and the background
($F_{background}$).

$$E = F_{sample} - F_{background} \ (Eq.\ 3),\ where$$

$$F = \frac{^{13}C}{^{13}C + ^{12}C} = \frac{R}{R+1} \ (Eq.\ 4),$$


$$And\ R = \frac{d^{13}C}{1000+1} \times R_{VPDB} \ (Eq.\ 5).$$

Subsequently, incorporation into bacterial biomass was calculated based on Middelburg et al. (2000) as:

$$I = \frac{sum\ I_{PLFA}}{a \times b} \qquad (Eq.\ 6)$$

where a is the average PLFA concentration in bacteria (0.073 g PLFA C g$^{-1}$ of C biomass in oxidised sediments (Brinch-Iversen and King, 1990) and b is the fraction of the bacterial PLFA considered here that are encountered in sediments of HAUSGARTEN (0.14; calculated from the fraction of i15:0 and ai15:0 in the control sediments
of Guilini et al. (2010) and those in the present study, data not shown). The total amount of algal C ($^{12}$C + $^{13}$C) recovered from bacteria, fauna, and DIC pools was calculated as the quotient of the total accumulation I and the fractional abundance of C in the algae (0.31 – 0.33).

## 3 Results

### 3.1 Organic carbon and nitrogen pools

The sediment at the study site was muddy (median grain size: 22 ± 4 µm), with an average organic carbon content of 1.03 dry wt% and nitrogen content of 0.12 dry wt% in the upper centimetre. Taking into account a porosity of 0.7 in the upper centimetre, this results in a TOC pool of 83 g $C_{org}$ m$^{-2}$ and a TN pool of 10 g N m$^{-2}$. Bacterial biomass made up 95-98 % (0.5 – 2.29 g $C_{org}$ m$^{-2}$, equalling 47-202 mmol $C_{org}$ m$^{-2}$) of the total benthic biomass (not including meiofauna <250 µm). Infauna (>250 µm) contributed with 0.028 – 0.054 g $C_{org}$ m$^{-2}$ (2.3-4.5 mmol
$C_{org}$ m$^{-2}$) and 0.005-0.025 g N m$^{-2}$ (0.4-1.8 mmol N m$^{-2}$; 2-5% of total benthic biomass). In terms of carbon biomass, infauna was dominated by polychaetes (44-75 %), calcareous foraminifera of the genus *Pyrgo* (12-21 %), with significant additional contributions of bivalves (1-10 %) and nematodes (2-4 %). In terms of nitrogen biomass, patterns where similar with polychaetes again dominating (55-84 %).



### 3.2 Algae addition

The addition of 69 mmol $C_{org}$ m$^{-2}$ *Thalassiosira* detritus and 37 mmol $C_{org}$ m$^{-2}$ *Emiliania* detritus reflects an addition of 1.0 % (*Thalassiosira*) and 0.5 % (*Emiliania*) fresh organic carbon to the background TOC pool in the upper centimetre of the sediment. For nitrogen, this corresponds to a 0.4 % (*Thalassiosira*) and 0.3 % (*Emiliania*) addition to the background pool.

Chlorophyll-*a* (chl-*a*) concentrations in the control sediments were low (1.4 µg ml$^{-1}$ in the 0-1 cm of the 4 d
experiment and 0.6 µg ml$^{-1}$ in the 14 d experiment). The addition of the fresh OM was reflected in an excess in chl-*a* in the surface sediments (Fig. S1). This excess was especially pronounced in the *Thalassiosira* experiments where the addition of phytodetritus more than doubled the chl *a* content in the uppermost cm as compared to the control. This increase is clearly associated with the algae addition as it largely exceeds the natural variability among replicates of surface sediments at the same station, which is on the order 15 % (Schewe, 2018).

### 3.3 Sediment POC and PN pools

After 4 days, labelled *Emiliania* and *Thalassiosira* biomass had accumulated in the organic carbon and nitrogen pools of the upper two centimetres of the sediment (Fig. 1 A, B). Accumulation of label continued in the 14 d experiment (Fig. 1 A, B). Over the time course, the subsurface sediment layers (1-3 cm) became increasingly enriched with carbon and nitrogen. Integrated over depth, the algae-derived matter that accumulated in the
sediments represented 30-60 % of the originally added *Thalassiosira* $C_{org}$ and 8-52 % of the originally added *Emiliania* $C_{org}$ (Table 1). In terms of nitrogen, the accumulation represented 36-137 % of the originally added *Thalassiosira* N and 14-65 % of the originally added *Emiliania* N.

### 3.4 Pore water labelled DIC and DIN pools

Respiration of $^{13}$C-labelled algae released $^{13}$C-DIC to pore waters in all four experiments, with higher amounts
found in the longer-lasting experiments (Fig. 1 C). The highest pore water $^{13}$C-enrichment was found in the 14 d *Thalassiosira* experiment concurrent with a strong vertical gradient in $^{13}$C-DIC. In the 14 d *Emiliania* experiment, on the other hand, a considerable share of the excess $^{13}$C-DIC was found in the deepest horizon (Fig. 1 C). Integrated over the sampled sediment depth, the algae-derived DIC made up ~0.1 % of the added organic carbon in both 4 d experiments and increased 10-fold to ~1 % of the added organic carbon in both 14 d experiments (Table
345    1).

Ammonium release from the algae to the pore waters, followed a similar pattern as $^{13}$C-DIC (Fig. 1 D). This release added considerably to the background ammonium pore water pool. At the end of the 4 d and 14 d *Thalassiosira* experiments, $^{15}$N-NH$_4^+$ concentrations represented 14-47% of the total ammonium found in the 0-1 cm sediment layer (compare with Fig. S2). At the end of the *Emiliania* experiments, $^{15}$N-NH$_4^+$ in the pore water of the 0-1 cm
sediment layer made up 4-18 % of the total ammonium concentration (compare with Fig. S2).

At concentrations that were approx. one order of magnitude lower compared to $^{15}$N-NH$_4^+$, also $^{15}$N-NO$_x^-$ accumulated in the pore waters of the experiments. In the surface (0-1 cm) sediment layer, these $^{15}$N-NO$_x^-$ concentrations made up < 1% of the total NO$_x^-$ pool. $^{15}$N-NO$_x^-$ concentrations decreased with depth in all experiments (Fig. 1 E).





In total, 0.2-0.6 % of the added algal nitrogen was recovered as DIN in the pore water of the 4 d *Thalassiosira* and *Emiliania* experiments. Algal-derived DIN continued to accumulate in the pore water to 1 and 4 % at the end of the 14 d *Emiliania* and *Thalassiosira* experiments, respectively (Table 1).

### 3.5 Labelled DIC dynamics in the overlying water

Dynamics of dissolved inorganic carbon and nitrogen species in the overlying water reflect the release from the sediment and are presented as a measure of the respiration and mineralization of the organic material by the sediment community. They may, however, to some extend also represent processes that take place within the overlying water itself.

In the first 3 days of both *Emiliania* incubations, $^{13}$C-DIC concentrations quickly accumulated in the overlying water, after which they levelled off (Fig. 2 A). The of $^{13}$C-DIC in the *Thalassiosira* chambers was more gradual.
The linear increases over time correspond to $^{13}$C-DIC fluxes in *Thalassiosira* chambers of 54 µmol m$^{-2}$ d$^{-1}$ (4d) and 124 µmol m$^{-2}$ d$^{-1}$ (14d) and in *Emiliania* chambers of 103 µmol m$^{-2}$ d$^{-1}$ (4d) and 18 µmol m$^{-2}$ d$^{-1}$ (14d, this linear increase was only significant at $p = 0.08$). Converted into algae-derived organic carbon (divide the $^{13}$C-DIC fluxes by fractional abundance of C in the algae, *i.e.* by 0.33 for *Thalassiosira* and by 0.31 for *Emiliania*), this results in a mineralization of 1-8 % of the added *Thalassiosira*-C$_{org}$ and 2-4 % of the added *Emiliania*-C$_{org}$ (Table
1) over the time-course of the respective deployments.

### 3.6 Total oxygen uptake

TOU in the control chambers as calculated from oxygen recordings of the optodes over time ranged between 0.4 mmol O$_2$ m$^{-2}$ d$^{-1}$ (14d) and 1.03 mmol O$_2$ m$^{-2}$ d$^{-1}$ (4d). The latter value is probably overestimating TOU owing to the presence of a shrimp in this particular benthic chamber. Due to the large range of oxygen uptake rates in the
control chambers and also strong variabilities in the treatments, the effect of the addition of fresh phytodetritus on TOU is not easily recognized: TOU in the *Thalassiosira* chambers ranged between 0.63 mmol O$_2$ m$^{-2}$ d$^{-1}$ (4d ) and 1.62 mmol O$_2$ m$^{-2}$ d$^{-1}$ (14 d), whereas TOU in the *Emiliania* chambers ranged between 0.14 mmol O$_2$ m$^{-2}$ d$^{-1}$ (4 d) and 0.46 mmol O$_2$ m$^{-2}$ d$^{-1}$ (14 d). Not including the exceptionally low TOU value in the *Emiliania* 4 d experiment, these fluxes are within the natural variability of the TOU measured for this area, which is in the range of 0.7 - 1.5
mmol m$^{-2}$ d$^{-1}$ (Wenzhöfer et al. unpublished data).

### 3.7 Labelled DIN dynamics in the water column

$^{15}$N-NH$_4^+$ concentrations in the overlying water of the *Thalassiosira* experiments initially increased (first 2.5 days: 7.7 and 4.3 µmol m$^{-2}$ d$^{-1}$ $^{15}$N-NH$_4^+$ in the 4 and 14 d experiments, respectively), before they started to decrease at relatively high rates (15.9 and 1.7 µmol m$^{-2}$ d$^{-1}$ $^{15}$N-NH$_4^+$ in the 4 and 14 d experiments, respectively). The $^{15}$N-
NH$_4^+$ concentrations in the overlying water were one order of magnitude higher than the $^{15}$N-NO$_x^-$ concentrations (Fig. 2 B). $^{15}$N-NO$_x^-$ concentrations steadily increased over time in both *Thalassiosira* experiments (0.56 and 0.35 µmol m$^{-2}$ d$^{-1}$ $^{15}$N-NO$_x^-$ in the 4 and 14 d experiments, respectively), which suggests continuous nitrification of ammonium released from *Thalassiosira* phytodetritus.



Different temporal patterns were observed in the *Emiliania* experiments. $^{15}$N-NH$_4^+$ concentrations in the overlying
water of the *Emiliania* 14 d experiment steadily decreased at a rate of 0.9 µmol m$^{-2}$ d$^{-1}$, while $^{15}$N-NO$_x^-$
concentrations increased at a rate of 0.38 µmol m$^{-2}$ d$^{-1}$.

Converted into algae-derived nitrogen (division by fractional abundance of 0.35 for *Thalassiosira* and by 0.43 for
*Emiliania*), the sum of the NH$_4^+$ and NO$_x^-$ fluxes results in a mineralization of 4-11% of the added *Thalassiosira*-
N and 0-2% of the added *Emiliania*-N (Table 1).

**3.8 Biotic response**

### 3.8.1 Bacterial numbers, activity and microbial community structure

Bacterial numbers as determined by means of AODC direct counts did not increase with the addition of fresh OM
(Fig. S4), but exo-enzymatic activities suggest that bacterial esterase activity in the upper two centimetres of
sediment increased by 19-36 % in the 4 d experiment (Fig. S5) as compared to control bacterial enzymatic activity
(1-1.55 nmol ml$^{-1}$ h$^{-1}$ in the upper two centimetre). This excess slightly exceeds the natural variability among
replicates, which is 15-17 % for surface sediments at the same station (Schewe, 2018). Bacterial enzymatic activity
was not measured in the 14 d experiment. Proteobacteria, in particular *Gammaproteobacteria*, dominated bacterial
communities in all samples (about 35 % of total community), but *Gammaproteobacteria* had a strikingly higher
proportional abundance (~ 60 %) in the *Thalassiosira* 14 d experiment (Fig. S6, S7), resulting in lower diversity
estimates based on the inverse Simpson index (Table S1). More specifically, *Colwelliaceae* and
*Oceanospirillaceae* were conspicuous families in sediments from this treatment, both accounting for about 15 %
of the total microbial community, in comparison to < 2 % in all other samples. The number of OTUs in the different
samples ranged between 700 and 1000, with highest numbers in the *Emiliania* 14 d treatment (Table S1).

### 3.8.2 Accumulation of algal carbon in bacterial biomass

Assimilation of the added fresh organic carbon into bacterial biomass determined via $^{13}$C-uptake into bacteria-
specific branched FAs (i – and ai 15:0) was highest in the surface sediments and increased over time (Fig. 3). With
time, also the subsurface bacteria incorporated the $^{13}$C-labelled carbon derived from the phytodetritus. Integrated
for the full sediment column investigated, bacterial carbon assimilation ranged from 0.3 % (*Emiliania*) to 1.2 %
(*Thalassiosira*) of the added organic carbon in the 4 d experiments and increased to 1.6 % (*Emiliania*) and 5.7 %
(*Thalassiosira*) of the added organic carbon in the 14 d experiments (Table 1).

### 3.8.3 Accumulation of algal carbon and nitrogen in infaunal biomass

Carbon assimilation by infauna >250 µm was similar in both 4 d experiments (0.03 mmol $^{13}$C$_{org}$ m$^{-2}$) and increased
over time, with the highest assimilation in the *Thalassiosira* 14 d experiment (0.13 mmol $^{13}$C$_{org}$ m$^{-2}$) (Fig. 4). The
nitrogen assimilation also increased over time, but the patterns were different, with a higher assimilation in the
*Thalassiosira* 4 d experiment than in the *Emiliania* 4 d experiment and the strongest assimilation in the *Emiliania*
14 d experiment (0.023 mmol $^{15}$N m$^{-2}$).

Foraminifera clearly dominated the carbon and nitrogen assimilation, followed by polychaetes. Foraminifera also
displayed the highest carbon- and nitrogen-specific assimilation of algal detritus, at least one (carbon-specific) to
two (nitrogen-specific) orders of magnitude lower than for polychaetes (Fig. 5). The specific uptake increased with





time in these two taxa, but was most pronounced in Foraminifera. Foraminifera displayed a higher carbon- and nitrogen specific uptake of *Emiliania* phytodetritus than of *Thalassiosira*. Bivalvia and Nematoda, the other two relatively abundant groups, hardly incorporated any phytodetritus-derived carbon or nitrogen into their tissue. In total, the infauna incorporated 0.6-0.8 % of the added carbon in the 14 d experiments, but up to 1.4% and 2.9 % of the added nitrogen in the *Thalassiosira* and *Emiliania* 14 d experiment, respectively (Table 1).

### 3.9 C and N budgets

#### 3.9.1 Carbon

In the 4 d *Thalassiosira* experiment, 2 % of the added algal organic carbon was processed (Table 1). A little less than half of this (43 %) was respired, half (50 %) was assimilated by bacteria, and 7 % had been assimilated by infauna (bacterial assimilation to infauna assimilation ratio: 7). After 14 d, 15.5 % of the added algal organic carbon had been processed, of which now two thirds (59 %) had been respired. Consequently the share of the added material that was assimilated by bacteria (37 %) and infauna (4 %) had decreased by this time (bacterial assimilation to infauna assimilation ratio: 9) (Table 1; Fig. 6).

Different patterns were observed in the *Emiliania* experiments. In the 4 d *Emiliania* experiment, 4 % of the added algal organic carbon was processed and distributed over the different pools. Most of the processed carbon had been respired to DIC (86 %), 7 % had ended up in infauna, and the same amount had been assimilated by bacteria (bacterial assimilation to infauna assimilation ratio: 1). After 14 d, still only 5 % of the added algal organic carbon had been processed but the distribution was different. A larger fraction of the processed organic carbon was present in bacterial (29 %) and infaunal (14 %) biomass (bacterial assimilation to infauna assimilation ratio: 2) and the portion respired amounted to 56 %.

#### 3.9.2 Nitrogen

In the 4 d *Thalassiosira* experiment, 11% of the added algal nitrogen was processed (Table 1). However, as biotic nitrogen assimilation only considers infauna and not nitrogen use by bacteria, the nitrogen budget presented should be regarded conservative. Most of the processed nitrogen was mineralized (88 %) and found back as ammonium and $NO_x^-$ in the water column or in the pore water DIN pool (5 %) and only 7% in infauna. After 14 d, 16 % of the added algal nitrogen had been processed, of which still only 9 % was traced back into infauna and the rest was dissolved in the overlying water pool and pore water pool (91 %) (Table 1; Fig. 6).

In the 4 d *Emiliania* experiment, only 0.6 % of the added algal nitrogen was processed. In contrast to organic carbon, a much larger share of organic nitrogen provided was assimilated. Three quarters were found back in infauna (73 %) and the rest (27 %) in the pore water DIN pool, whereas no significant DIN release to the overlying water column was observed. After 14 d, 6 % of the added algal nitrogen had been processed, and still the fauna share was high (49 %) and the rest was mainly found back as DIN in the overlying water column and pore water pool (51 %).

In all experiments, still a considerable amount (8-60 % of the added algal organic carbon; 14-82 % of the added algal N) of uncharacterized OM (bulk sedimentary $^{13}C$ POC and $^{15}N$ PON) was left in the sediment (Table 1). This pattern was most clear in the *Thalassiosira* experiments. A large fraction could not be recovered: in particular in



the 4 d experiments, 68-88 % of the added algal organic carbon and 7-86 % of the added algal N was missing, hence carbon and nitrogen budgets cannot be closed. An exception to this pattern is the nitrogen budget of *Thalassiosira* 14 d, in which the amount of uncharacterized organic nitrogen even exceeds the total amount of added algal nitrogen (Table 1).

**4 Discussion**

We hypothesised that a potential climate change-related shift in phytoplankton communities from diatoms to coccolithophorids would have implications for bathyal benthic OM assimilation and affect the mineralization patterns at the deep-sea floor. Our results indeed show shifts in the importance of infaunal *vs.* bacterial assimilation of algal organic carbon with a lower bacterial/infaunal assimilation ratio in *Emiliania* treatments compared to
*Thalassiosira* . In addition, both the cycling pathways of organic carbon and nitrogen point at a less efficient mineralization of *Emiliania* detritus as compared to *Thalassiosira* detritus: after 14 days, 5 times less carbon and 3.8 times less nitrogen of the *Emiliania* detritus was recycled . This indicates that the cycling of *Thalassiosira* detritus was faster compared to *Emiliania*.

The added *Thalassiosira* and *Emiliania* doses differed by 35% in terms of total carbon (46% in terms of organic
carbon) because of methodological complications. The above-mentioned difference in OM recycling between both phytoplankton species therefore could also be partly driven by food quantity. Experimental studies on the effect of resource quantity on benthic mineralization pathways indeed point at a 2-10-fold increase in bacterial carbon assimilation and 6-36 times higher carbon respiration with a 10-fold increase in OM dose (Bühring et al., 2006; Gontikaki et al., 2013; Mayor et al., 2012; van Nugteren et al., 2009). However, the difference in added OM
between the experimental treatments of these earlier studies was one order of magnitude, or more, which is at least five times larger than the difference in dose between the *Thalassiosira* and *Emiliania* treatments in this study. Modelling also suggested that, in case of food-quantity driven alterations in OM degradation patterns, it is rather a high POC input to the abyss that results in a stronger role for fauna mediated carbon cycling (Dunlop et al., 2016), whereas we found a smaller role for fauna in the treatment with the highest POC input (*Thalassiosira*).
Based on both the above-mentioned experimental and modelling studies, we do not expect large effects of the relatively small differences in the quantity of the added *Thalassiosira* and *Emiliania*.

**4.1 Carbon respiration**

The relatively low amount of algal carbon mineralized in this study (with a maximum of 15.5 % of *Thalassiosira* biomass after 14 d) and the relatively slow response to OM addition is similar to other deep-sea studies where low
temperatures and limited biomass slow down the recycling of OM (Witte et al., 2003a; Woulds et al., 2009).

Similar to other cold deep-sea sediments (e.g. Woulds et al., 2009), the fresh phytodetritus was mainly respired, *i.e.* traced back in DIC in the overlying water or pore water pools, whereas the assimilation into biomass was less important. In accordance with Soltwedel et al.(2000), bacteria clearly displayed the highest biomass in our study area (95 % of the assessed biotic organic carbon pool), whereas infauna biomass never exceeded 5 % of the total
biomass. Therefore, the respiration of the added algae is, most likely, primarily attributed to bacteria. This corroborates previous experimental studies (Moodley et al., 2002; Witte et al., 2003c) and is in agreement with





earlier *in situ* measurements (Donis et al., 2016) and food web modelling results that assigned 93 % of the respiration to bacteria (van Oevelen et al., 2011). An increase in bacterial activity was clearly observed after the deposition of fresh phytodetritus, both in terms of respiration and enzymatic activity, which is in accordance with
earlier studies (Boetius and Lochte, 1994; Hoffmann et al., 2017). Also, bacterial fatty acids indicate an uptake of algal-derived carbon, while  no significant increase in abundances was observed even after 14 d (Fig. S4). This may be related to the relatively short experimental time frame and slow turnover time of microbial communities in Arctic deep-sea environments - of the order of 4-5 weeks (Boetius and Lochte, 1994; Hoffmann et al., 2017).

Overall, bacterial community composition was similar to previous reports from global and Arctic deep-sea
sediments (Bienhold et al., 2016; Hoffmann et al., 2017). The changes in community composition in response to the addition of phytodetritus were partly consistent with previous studies from the same area (Hoffmann et al., 2017), e.g. the relative increase in *Colwelliaceae* (*Gammaproteobacteria*) in the *Thalassiosira* 14 d experiment. In contrast, there seemed to be little increase in *Bacteroidetes*/*Flavobacteria* which are usually typical degraders of complex algal organic material (Hoffmann et al., 2017; Teeling et al., 2012). However, in the current
experiment, a clear response (*i.e.* change in community composition) was only observed for the *Thalassiosira* 14 d treatment. This may be a consequence of the less efficient use of *Emiliania* OM and the lower contribution of bacteria in the assimilation of carbon in these treatments, especially considering the relatively short experimental time frame and the above-mentioned slow doubling times for bacterial communities in Arctic deep-sea sediments.

It seems that *Emiliania* OM was initially (4 d and start of 14 d experiment) more respired than *Thalassiosira* (in 4
d experiment: 4 % of the added *Emiliania* OM, of which 3.6 % by DIC release, as opposed to 2 % of the added *Thalassiosira* OM), but this could as well be ascribed to dissolution of the inorganic coccoliths. The absence of observable $NH_4^+$ or $NO_x^-$ release in the *Emiliania* 4 d experiment agrees with this view.

**4.2 Nitrogen cycling**

Mineralization of the phytodetritus was also observed in terms of nitrogen. In deep-sea sediments with low OM
content and deep oxygen penetration, nitrate is expected to be the dominating form of nitrogen recycled (Brunnegård et al., 2004). Measurements of nitrogen cycling in oligotrophic deep-sea environments are very scarce (Berelson et al., 1990; Brunnegård et al., 2004) and to our knowledge do not exist for Arctic deep-sea sediments. Most of the OM is aerobically mineralized in the upper centimetre of the sediment in our study area (Donis et al., 2016). Hence we assume that denitrification of nitrate does not occur in the oxidized sediment layer of our
experiments and that the observed accumulation of nitrate in the overlying water is caused solely by nitrification. In this case, the nitrification rates derived from turnover of the algal $^{15}N$ equal 0.35 and 0.38 µmol $^{15}N$ m$^{-2}$ d$^{-1}$ (*Emiliania* and *Thalassiosira* 14 d experiments respectively) and 0.56 µmol $^{15}N$ m$^{-2}$ d$^{-1}$ (*Thalassiosira* 4 d experiment). To express these rates in terms of algal N nitrified, the labelling percentage of the total $NH_4^+$ concentrations has to be taken into account. Because $^{15}N$-$NH_4^+$ was produced during the release from the algal
detritus, the labelling fraction ($^{15}N$-$NH_4^+$: total $NH_4^+$) in the pore water increases exponentially (Song et al., 2016). This can be observed in the upper cm of the sediment where the labelling fraction increased over time (compare Fig. 1 D with Fig. S3; calculated labelling fraction in Table S2). Taking these labelling fractions of $^{15}N$-$NH_4^+$ into account, the according nitrification rates are then 0.74 µmol N m$^{-2}$ d$^{-1}$ (*Thalassiosira* 14 d experiment), 2.1 N m$^{-2}$ d$^{-1}$ (*Emiliania* 14 d experiment) and 4.0 µmol N m$^{-2}$ d$^{-1}$ (*Thalassiosira* 4 d experiment), an order of magnitude



lower than the nitrification rates observed in other oligotrophic deep-sea areas (Berelson et al., 1990; Brunnegård et al., 2004). Altogether, only 0.4-1.5 % of the total nitrification would then be attributable to nitrification of the ammonium released by the algal detritus. However, these nitrification estimates are based on an addition of fresh phytodetritus that was heavily (99 %) diluted into the (labile + refractory) OM pool of the sediment, hence an underestimation.

**4.3 Assimilation into biomass**

Altogether, 0.6-6.3 % of the added carbon was assimilated into biomass. Within this fraction, bacteria seemed to be key players in cycling of freshly added *Thalassiosira* phytodetritus (~90 % of the biological assimilation occurred through bacteria, ~10 % through infauna). The share of bacteria was smaller in the *Emiliania* experiments (48-68 % of the biotic assimilation is accounted for by bacteria, 32-52 % by infauna). This difference is not driven
by a difference in biomass since the four experiments displayed a similar bacterial *vs.* infauna biomass ratio. The absolute assimilation by infauna was similar in all experiments, whereas the absolute assimilation by bacteria was ~7 times higher in the *Thalassiosira* compared to the *Emiliania* experiments. This seems to indicate that diatom biomass was more easily accessible to bacteria as compared to coccolithophorids. This is in contrast with water column studies of Iversen and Ploug (2010), who found that the degradability of phytoplankton in surface waters
depended partly on the structure of the external mineral protection. In the silica diatom frustrule, Si–C or Si–O–C interactions are thought to protect silica from dissolution until the organic matrix is removed by bacteria (Moriceau et al., 2009 and references therein). Similarly, the calcite matrix of the coccoliths can act as a physical barrier against bacterial degradation (Engel et al., 2009). However, comparable carbon-specific respiration rates were measured for aggregates of *Emiliania* and *Skeletonema* diatoms, suggesting similar degradability in surface waters
(Iversen and Ploug, 2010). Increased hydrostatic pressure such as in the deep-sea, leads to faster dissolution of the coccoliths than in surface waters, but in the presence of natural prokaryotic communities also induces more aggregation (Riou et al., 2018), which can offer again organic matter protection from solubilisation and remineralization (Engel et al., 2009). On the other hand, silicate dissolution is reduced when diatoms embedded in sinking aggregates fall through the water column (Tamburini et al., 2006). It is therefore unclear whether diatoms
should be easier to degrade by bacteria than coccolithophores when they reach the deep-sea floor. Our data suggest that in Arctic deep-sea sediments, *Thalassiosira* might be more easily degraded by bacteria than *Emiliania*. Nevertheless, the state in which the two species were added at the start of our experiments, might differ from the aggregates and faecal pellets formed during the descent through the water column.

Also noteworthy is the fact that larger organisms dominated the competition for fresh food: bacterial carbon
assimilation never exceeded 1 % of bacterial biomass, while infauna carbon assimilation reached on average 10 % of their biomass, in Foraminifera even 40% of their carbon biomass (Fig. 5).This agrees with other studies showing that bacteria dominate fresh OM assimilation in sediments only if they are almost devoid of fauna (Moodley et al., 2005; van Oevelen et al., 2011). However, in systems where fauna is present, the latter rapidly consume the fresh phytodetritus (Blair et al., 1996; Levin et al., 1997, 1999; Moodley et al., 2002, 2005; Witte et
al., 2003a, 2003b). This biased ratio indicates that bacteria may be initially outcompeted by infauna or that carbon becomes available to bacteria only after it has passed through the guts of fauna as has been hypothesized earlier (Witte et al., 2003a, 2003b). However, even after 14 days, Foraminifera still had a two orders of magnitude higher carbon-specific assimilation than bacteria, implying that larger organisms continued to dominate the competition





for fresh OM. This confirms earlier studies showing that foraminifera can be key players in the early diagenesis of fresh OM at the deep-sea floor (Moodley et al., 2000, 2002; Nomaki et al., 2005; Woulds et al., 2007).

Although macrofaunal (>250µm) foraminifera can have a retarded response to phytodetritus inputs as compared to smaller (> 63 µm) foraminifera (Sweetman et al., 2009), the carbon assimilation rate by macrofaunal foraminifera in this study is similar to that of smaller foraminifera at Station M (Enge et al., 2011). Foraminifera also seemed to prefer *Emiliania* over *Thalassiosira*, as seen in the carbon and nitrogen-specific assimilation. This
is in contrast with observations of abyssal foraminifera at station M that showed a preference for nitrogen from diatoms over nitrogen from coccolithophores (Jeffreys et al., 2013). However, deep-sea foraminifera demonstrate a variety of dietary preferences (Gooday et al., 2008), and the species found in this study differ from the ones found in Jeffreys et al. (2013).

### 4.4 C:N preference

Measuring the degradation pathways of phytodetritus that is both $^{13}$C and $^{15}$N labelled allows for comparing the preferential assimilation in those pools where both isotopes could be traced, *i.e.*, processed pools (respiration in overlying water, pore water), assimilation by infauna (assimilation of N into bacterial biomass could not be addressed based on the measurements performed in this study), and 'leftovers' (sediment POC). The C:N ratio of the added phytodetritus was similar for both types of algal detritus, with *Thalassiosira* (13.3) being a little poorer
in nitrogen as compared to *Emiliania* (12.6). The molar C:N ratio in all processed pools of the short term experiments was highest in the *Emiliania* 4 d experiment (on average 14.3; Table 2) and considerably lower in the *Thalassiosira* 4 d experiment (on average 3.7). This suggests that in the first stages after the addition of *Thalassiosira*, nitrogen is preferably used. This seems to be in agreement with the nature of the POC leftovers in the sediment, which were higher in carbon content than the processed pools (C:N ratio 8.7). In the *Emiliania* 4 d
experiment on the other hand, the preferred use of nitrogen might be masked by the dissolution of the carbonates from the coccoliths, leading to high carbon content in overlying and pore water. The remaining OM therefore contains less carbon, hence the lower C:N ratio compared to the processed pools. After 14 days, the C:N ratios of both processed and unprocessed pools in both experiments became increasingly enriched in carbon (higher C:N ratio). This points again to carbon being increasingly processed only after the nitrogen content of the available
phytodetritus had strongly been consumed. An exception to these patterns is the C:N ratio of the infauna in the *Emiliania* experiments that decreased over time. Infauna feeding on *Emiliania* also ingest the coccoliths. As also observed in the absolute carbon and nitrogen assimilation, it seems that it takes longer to release the OM from the coccoliths into the guts of infauna, hence the nitrogen was only accessible later.

### 4.5 Carbon and nitrogen inventories

The carbon and nitrogen budgets were not closed (Fig. S8). Unclosed budgets are common (Buhring et al., 2006; Evrard et al., 2012; Middelburg et al., 2000; Witte et al., 2003a; Woulds et al., 2009) and can be partly attributed to methodological causes:

(1) We missed certain pools: (a) We could not sample the sediment subsurface layers of the 4 d *Emiliania* experiment as deeper layers where lost upon retrieval of samples from the chambers on board, and as such miss
the subsurface processing pathway. However, in the other experiments, this part accounts for < 10 % of the sum



of the total processed carbon and uncharacterized OM. (b) Unmeasured parts of the carbon and nitrogen budgets are the DOC and DON pools in the overlying water and pore water. A modelling study in the same area estimated that > 25 % of the total carbon input to the food web quickly dissolves into DOC that is taken up and then respired by prokaryotes (van Oevelen et al., 2011). Based on this finding one would expect that the released DOM in the sediment – at some point but not necessarily in the time frame of the experiments – reappears in the bacterial biomass or respired pools. (c) We did not consider meiobenthos < 250 µm, since nematodes, the most abundant metazoan component of deep-sea meiobenthos, are usually responsible for only < 1 % of the total mineralization (Ingels et al., 2010).

(2) We added less than we assumed: A substantial part (8-60 %) of the carbon budget concerns uncharacterized particulate OM in the sediment. The share of this OM increases from 4 d to 14 d, which makes the 14 d budgets more closed. This increase cannot be attributed to a difference in the amount of algal matter initially added. It may indicate, that settlement of the algal OM to the sediment surface has taken several days and that not the full amount has been available to the sediment community right from the start. However, we did not observe increased amounts of POM in the unfiltered samples of the overlying water. Nevertheless, it cannot be ruled out, that parts of the added OM disintegrated into colloidal particles or was released from the cells as DOM that took longer to arrive at the sediment. (3) The added phytodetritus might not have been distributed evenly across each benthic chamber. This means that the location within the chamber from which samples for bacterial assimilation and tracer in the pore water were taken will have affected how much C and N was found in faunal and bacterial biomass. (4) Finally, the missing part of the budgets and the excess in the 14 d *Thalassiosira* nitrogen budget might be a result of accumulated errors in integration procedures, due to spatial variability or uncertainties in conversion factors (Middelburg et al., 2000). For these reasons, the nature of the missing carbon and nitrogen cannot be fully resolved.

### 4.6 Long term implications

Despite its constraints, the presented experiments describe a snapshot of the potential effects of a change in food quality for Arctic deep-sea benthos. What could be the long-term implications of a shift in the quality of food arriving at the deep-sea floor? Our observations suggest that the degradation of *Emiliania* - dominated phytodetritus could be less efficient than that of *Thalassiosira* - dominated detritus. If this is the case, this shift would affect the recycling of fresh phytodetritus and the regeneration of nutrients.

Also the food web structure could be altered: with a shift in food quality, the share of infauna in OM degradation could increase as compared to bacteria. An increased POC flux to the Arctic deep-sea floor because of reduced sea-ice cover could in parallel trigger a switch in dominance of benthic organic matter processing by bacteria to dominance by metazoans, with implications for the upper food-web levels (Sweetman et al., 2017, p.201). Arctic deep-sea communities could be flexible in their response to new food sources that accompany climate change and could likely also be influenced by changes in the *amount* rather than the type of OM reaching the bottom (Sun et al., 2007). At our study site however, there are signs from long term studies that both food quality and quantity affect the densities and trophic diversity of benthic communities (Soltwedel et al., 2015). This calls for additional *in situ* experiments in the Arctic deep-sea supplying the benthos with different quantities of OM. Also OM quality should receive further attention, since shifts in phytoplankton community structure in Fram Strait have now turned from diatom-coccolithophore dominated blooms to diatom – *Phaeocystis pouchetti* dominated blooms (Nöthig et





al., 2015; Soltwedel et al., 2015). *Phaeocystis* blooms have been shown to be low in food quality and may impede
development in some grazing species (Breteler and Koski, 2003; Tang et al., 2001). Future investigations on the
effects of altered phytodetritus input to Arctic deep-sea benthos should therefore also involve the fate of
*Phaeocystis pouchetti* blooms and their role for bentho-pelagic coupling in Fram Strait.

## 5 Data availability

All data are available at PANGAEA (https://doi.pangaea.de/10.1594/PANGAEA.885617).

## 6 Acknowledgements

We wish to thank the captain and the crew of RV Maria S Merian expedition MSM29 for their help during the
expedition to the LTER observatory HAUSGARTEN. We further would like to thank Anja Pappert for culturing
the algae, the MPI-SeaTech technicians for preparing and operating the lander, Christiane Hasemann for analyses
of FDA hydrolysis, Martina Alisch, Rafael Stiens and Gabriele Klockgether for laboratory analysis of sediment
properties, Jenny Wendt for help with PLFA extractions, Clara Martínez Pérez for infauna EA-IRMS analyses and
Thorsten Dittmar for DOM analyses. Katja Guilini and Nicolas Van Oostende are acknowledged for fruitful
discussions and Ann Vanreusel for critical reading of the manuscript. This work contributes to the framework of
the HGF Infrastructure Program FRAM of the Alfred-Wegener-Institute Helmholtz Center for Polar and Marine
Research. Funding was received from the European Research Council Advanced Investigator grant 294757, from
the Research Foundation - Flanders (FWO Belgium) to U.B. (grant nr. 1201716N).

## 7 Author contributions

Designed the experiment: F.W., F.J.; performed the experiment: F.J., Ch.B., F.W.; analysed the samples: U.B.,
Ch. B., H.M., M.E., Ca.B.; analysed the data: U.B., G.L., H.M., Ch.B.; wrote the manuscript: U.B. with
contributions from all authors.

## 8 Competing interests

The authors declare that they have no conflict of interest.





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





**Figures**

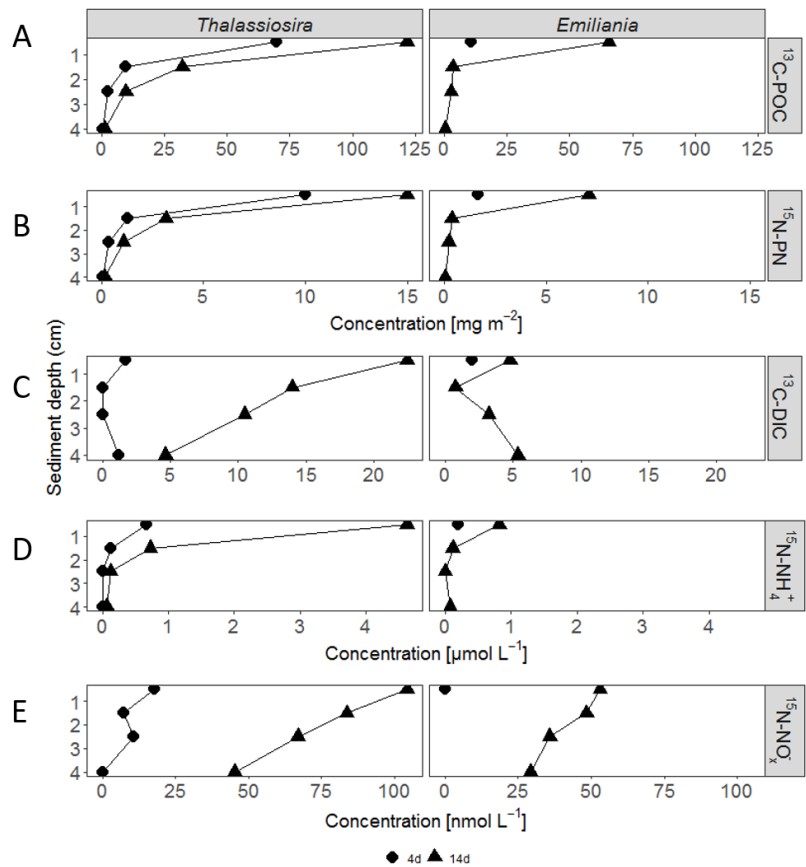

**Figure 1: Concentration of excess bulk $^{13}$C-POC (A) and $^{15}$N-PN (B) in the sediment and $^{13}$C –DIC (C), $^{15}$N-NH$_4^+$ (D) and $^{15}$N-NO$_x^-$ (E) in the pore water of each experiment. Only the first sediment cm could be sampled in the *Emiliania* 4 d experiment.**


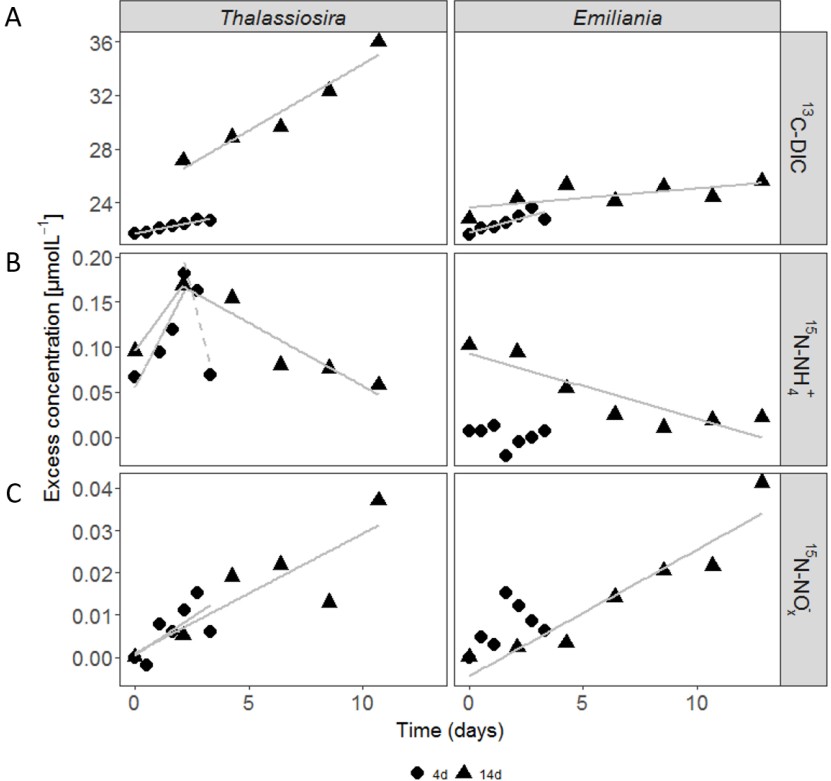

**Figure 2:** Accumulation of $^{13}$C-DIC (A), $^{15}$N-NH$_4^+$ (B) and $^{15}$N-NO$_x^-$ (C) in the water column over time in the experiments. Only significant linear regressions are shown in full grey lines. The dashed grey line in the $^{15}$N-NH$_4^+$ data of the *Thalassiosira* 4 d experiment represents a non-significant regression based on 3 points (p = 0.2).

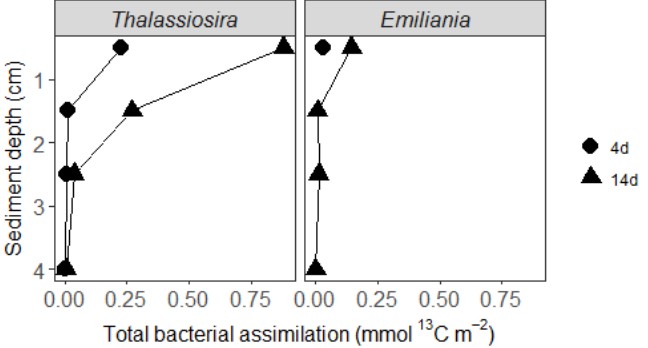

**Figure 3:** Total assimilation of algal carbon (mmol $^{13}$C m$^{-2}$) in bacterial biomass in the sediment of the 4 d and 14 d *Emiliania* and *Thalassiosira* experiments. For the 4 d *Emiliania* experiment, only the first sediment cm was available for analysis.





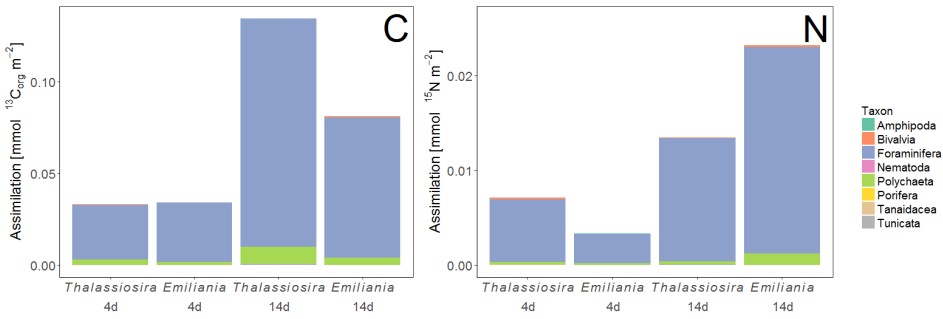

**Figure 4: Algal detritus assimilation by the different infauna taxa in terms of carbon (left) and nitrogen (right).**

**Only the first sediment cm could be sampled in the *Emiliania* 4 d experiment.**





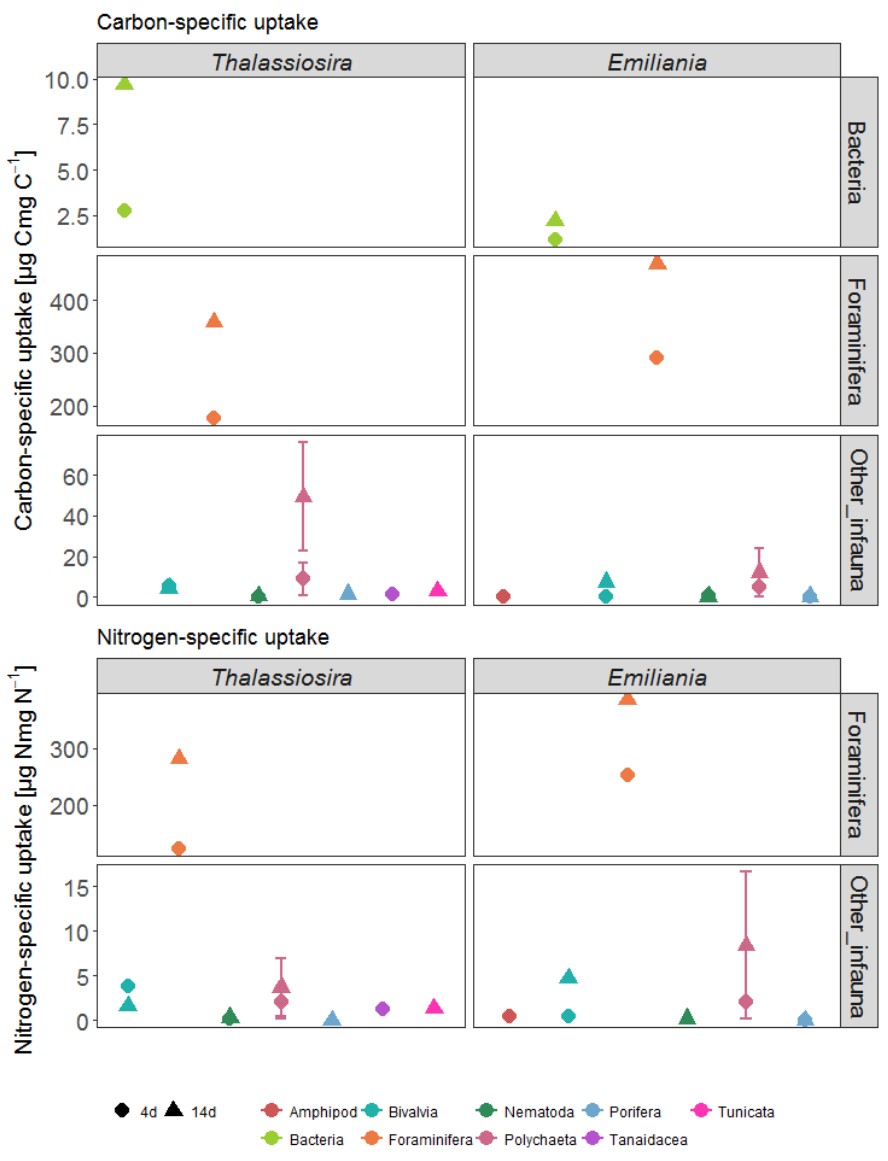

**Figure 5: Specific uptake of carbon and nitrogen for infauna > 250 µm and bacteria. No nitrogen-specific uptake was quantified for bacteria. Error bars indicate sd of specific uptake of replicate samples.**



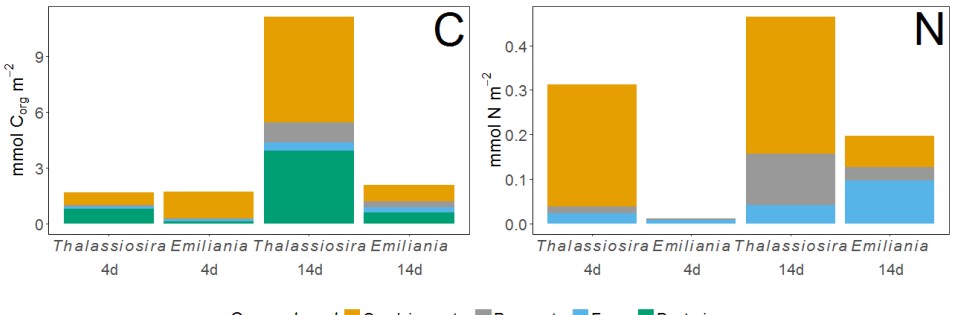

**Figure 6: Processed C$_{org}$ (left) and N (right) in the different compartments. Note that bacterial assimilation was not quantified in terms of N. Only the first sediment cm could be sampled in the *Emiliania* 4 d experiment.**






931  **Tables**

932  **Table 1: Carbon and nitrogen stock inventories for the four experiments. Data are shown as concentrations (mmol m⁻²) and**
933  **as a percentage of the amount of carbon or nitrogen added as algal detritus. *Only the first sediment cm could be sampled**
934  **in the *Emiliania* 4 d experiment.**

| | *Thalassiosira* 4 d | | | *Emiliania* 4 d* | | | *Thalassiosira* 14 d | | | *Emiliania* 14 d | | |
|---|---|---|---|---|---|---|---|---|---|---|---|---|
| | mmol $C_{org}$ m⁻² | % of total | % of processed | mmol $C_{org}$ m⁻² | % of total | % of processed | mmol $C_{org}$ m⁻² | % of total | % of processed | mmol $C_{org}$ m⁻² | % of total | % of processed |
| Infauna assimilation | 0.1 | 0.2 | 6.8 | 0.1 | 0.3 | 7.4 | 0.4 | 0.6 | 4.1 | 0.3 | 0.8 | 14.1 |
| Bacterial assimilation | 0.8 | 1.2 | 50.1 | 0.1 | 0.3 | 6.8 | 3.9 | 5.7 | 36.6 | 0.6 | 1.6 | 29.4 |
| Respiration (DIC) | 0.6 | 0.9 | 39.2 | 1.3 | 3.6 | 83.0 | 5.3 | 7.6 | 49.2 | 0.8 | 2.2 | 40.7 |
| Pore water DIC | 0.06 | 0.1 | 3.8 | 0.04 | 0.1 | 2.7 | 1.1 | 1.6 | 10.2 | 0.3 | 0.9 | 15.8 |
| POC leftover in sediment | 20.7 | 30.0 | | 2.9 | 7.7 | | 41.7 | 60.5 | | 19.4 | 52.1 | |
| Missing | 46.7 | 67.7 | | 32.7 | 88.0 | | 16.6 | 24.0 | | 15.8 | 42.5 | |
| Added at start | 69 | | | 37 | | | 69 | | | 37 | | |
| Total processed | **1.6** | **2.3** | | **1.6** | **4.3** | | **10.7** | **15.5** | | **2.01** | **5.4** | |
| | | | | | | | | | | | | |
| | mmol N m⁻² | % of total | % of processed | mmol N m⁻² | % of total | % of processed | mmol N m⁻² | % of total | % of processed | mmol N m⁻² | % of total | % of processed |
| Infauna assimilation | 0.02 | 0.4 | 7.0 | 0.01 | 0.4 | 72.7 | 0.04 | 1.4 | 8.8 | 0.06 | 2.9 | 48.6 |
| Bacterial assimilation | - | | - | - | | - | - | | - | - | | - |
| DIN mineralization | 0.3 | 4.4 | 87.8 | 0.0 | 0.0 | 0.0 | 0.3 | 10.7 | 66.4 | 0.04 | 2.1 | 35.5 |
| Pore water DIN | 0.02 | 0.6 | 5.2 | 0.003 | 0.2 | 27.3 | 0.1 | 4.0 | 24.8 | 0.02 | 1.0 | 16.0 |
| PN leftover in sediment | 2.4 | 35.6 | | 0.27 | 13.8 | | 4.0 | *136.6* | | 1.3 | 64.5 | |
| Missing | 0.2 | 59.4 | | 1.7 | 85.6 | | *-1.5* | *-52.8* | | 0.6 | 29.5 | |
| Added at start | 2.9 | | | 2 | | | 2.9 | | | 2 | | |
| Total processed | **0.31** | **10.8** | | **0.01** | **0.6** | | **0.46** | **16.1** | | **0.12** | **6.0** | |

935





936  **Table 2: Molar C:N ratios for each of the measured pools. *C:N ratio not calculated because N-mineralization was below**
937  **detection limit.**

|  | *Thalassiosira* 4 d | *Emiliania* 4 d | *Thalassiosira* 14 d | *Emiliania* 14 d |
|---|---|---|---|---|
| **Infauna assimilation** | 5 | 14.5 | 10.8 | 4.9 |
| **Respiration** | 2.3 | -* | 17.1 | 19.4 |
| **Pore water** | 3.8 | 14.1 | 9.4 | 16.7 |
| **POC leftover** | 8.7 | 10.4 | 10.5 | 15.1 |

938

939