# Peer review of "Carbon and nitrogen turnover in the Arctic deep sea: in situ benthic community response to diatom and coccolithophorid phytodetritus"

_Biogeosciences, 2018_

## Referee Comment (RC1) · J. Middelburg (Referee) · 21 Aug 2018

Braeckman and colleagues present the results of in situ experiment in the Arctic deep sea. Global warming is causing shifts in phytoplankton community composition and functioning, with diatoms being replaced by coccolithophores and other phytoplankton groups. This has consequences for the quantity and quality of organic matter exported and reaching the seafloor. The authors have used 13C and 15N labelled phytodetritus from a diatom and a coccolith to trace the (differential) fate and flows of C and N through the entire food web.

[Figure]

Major strengths of this paper are (1) the focus on Arctic deep-sea sediments (few studies so far), (2) the use of in situ techniques at 2500 m water depth, (3) the whole system, comprehensive approach. The authors did a decent job by tracing not only C, but also N, for following C and N not only through food-web compartments (remaining detritus, bacteria and fauna), but also in overlying water and pore-waters. For instance, they quantified nitrogen transfer from phytodetritus via the sedimentary organic matter and fauna pools to ammonium and nitrate in overlying water and pore waters; thus connecting elemental flow biogeochemistry and microbial transformations with food-web processing. In fact, this is among the most complete tracer recovery studies to date, at par with some recent coastal studies.

However, as clearly communicated by the authors, the level of replication is suboptimal. The study is based on two lander deployment, one lasting 4 days and another one lasting 14 days. Each deployment involved three chambers, one control, one diatom and one coccolith treatment. Sensu stricto there is no replication. In a world without logistic and financial constraints one would indeed prefer full replication as is now the standard in most in situ coastal studies and ship-board and laboratory studies. However, I do understand and value their approach given the resources available. The experimental design needs to balance the requirement of full replication for standard statistical inference testing on the one hand, with the need for at least some temporal dynamic information (e.g. there was no a priori knowledge about the optimal experimental duration) and the requirement of in situ experimentation to obtain unbiased transfer rates on the other hand. Overall, the benthic community responses after 4 and 14 days of incubation were rather similar (i.e. the basic findings were replicated and controls were similar) although details (e.g ratio assimilated vs. respired etc.) differed.

Specific remarks:

_ Line 37: 2.7-2.8 for clarity

_ Line 45: years phytoplankton blooms became more mixed

_ Line 59: delete [OM] because you introduce abbreviation again one line lower.

_ Line 83-89: The authors attribute potential differences in response to diatoms vs. coccolith phytodetritus almost entirely to differences in their skeletons (Si vs CaCO3) ignoring biochemical composition aspects.

_ Line 118: The TOC of algae can never be 78 or 95%, because 100 % organic matter corresponds to 40-50% C depending on biochemical make up. Please correct.

_ Line 122 & 125: TDN: is this indeed total dissolved nitrogen. But was the inorganic nitrogen not removed by washing three times, so that TDN is more or less DON?

_ Line 186: recovered or added labelled phytodetritus?

_ Line 267: Freeze-dried algae were measured for total 13C and 15N, thus for coccolith inorganic and organic carbon were combined Both types of carbon will very likely be similarly enriched given the identical carbon sources. What is unclear though is how much of the 13C-DIC attributed to respiration (and recovered in overlying water and pore-water) is from dissolution of the carbonate. Combining 15N-DIN and 13C-DIC release might be give some hints whether 13C from carbonate dissolution matters or not.

_ Line 290-296: I do not see the use of these equations. You present all your data in excess 13C atom fractions. Why then are equation 4, 5, etc needed (these are copy-pasted from prior work in which del values were reported). Line 289-296 can be deleted without loss of information.

_ All through try to avoid using on the other hand if there is no on the one hand (I have counted it 3 or 4 times).

_ Line 364:.. after which the increase levelled off. . . . . . The increase of 13C-DIC. . .was higher and steady.

_ Line 433: was respired (and recovered in overlying water and pore water). . . ..

_ Section 4.1: Cold, deep-sea systems sometimes show a delayed response, i.e. low activities during the first two-three days (e.g. Andersson et al 2008 in Arabian Sea). This is one of the reason why your experimental design (two incubation durations) makes sense. Culturing phytoplankton in the lab followed by freeze-drying before additions might perhaps have resulted in the addition of DOC to your experiment at the seafloor. This really depends on the very details of your phytodetritus preparation. Differences in response among studies in the literature can be partly explained by this. Resolution requires 13C measurement of the DOC pool and that is a daunting task.

_ Line 525: unclear, too cryptic, I understand but will all readers?

_ Line 536: 0.4-1.5% of total nitrification is similar to 0.3-0.4 contribution to sediment ON pool.

_ Line 570-572: an additional/alternative explanation. OM delivered to the sediment surface is far away from microbes in the subsurface. Animals, through their bioturbation, mix OM down and deliver OM to bacteria that do not move. This is another component of the (inverted) sediment microbial loop.

_ Line 588: sediment OM

_ Line 593: the OM leftovers (POC cannot have a C:N ratio).

_ Line 610-618: another reason for non-closure is that 15N in bacteria and 13C/15N in archaea have not been measured (although the latter might contribute just a minor amount).

_ Line 641: delete p. 201

_ Line 656: We thank. . .. . .. We further thank Anja. . .

_ References: balanced coverage, but I missed the Boetius et al. 2012 note in Science.

---

## Author Comment (AC1) · 7 Sep 2018

**Reply to review of Jack J Middelburg**

**Reviewer's comment:** Line 37: 2.7-2.8 for clarity

**Reply:** Nöthig et al. (2015) report 2.7-8 °C, not 2.7-2.8°C, so we leave the temperature range as originally stated.

**Reviewer's comment:** Line 45: years phytoplankton blooms became more mixed

**Reply**: Agreed, changed accordingly.

**Reviewer's comment:** Line 59: delete [OM] because you introduce abbreviation again one line lower.

**Reply:** We thank the reviewer for noticing. Corrected accordingly.

**Reviewer's comment:** Line 83-89: The authors attribute potential differences in response to diatoms vs. coccolith phytodetritus almost entirely to differences in their skeletons (Si vs $CaCO_3$) ignoring biochemical composition aspects.

**Reply:** The reviewer is correct. We added the biochemical composition aspects to lines 83-84 and 89-96:

"One mechanism would be the impediment of food source utilization by the physical protection of the cells."

[…]

"The coccolithophore *E. huxleyi* for example, contains comparatively high levels of n-3 polyunsaturated fatty acids, essential for growth and reproduction of eukaryotic consumers (Pond and Harris, 1996). This high nutritional value has been used to explain the higher survival rate of planktonic foraminifera (Anderson et al., 1979) and egg production by calanoid copepods (Neystgaard et al., 1997) as compared to when these organisms were fed a diatom diet."

**Reply (cont.)** In the discussion about the high foraminiferal preference for *Emiliania*, we also touched upon the biochemical composition again (lines 603-605):

"This agrees with the higher survival rate of planktonic foraminifera in feeding experiments with *Emiliania* than with diatoms (Anderson et al., 1979), which was later related to the higher nutritional value of *Emiliania* (Pond and Harris, 1996), …"

**Reviewer's comment:** Line 118: The TOC of algae can never be 78 or 95%, because 100 % organic matter corresponds to 40-50% C depending on biochemical make up. Please correct.

**Reply:** The reviewer is right. We corrected the sentence to:

"The corresponding TOC of the algae was 78 % of TC (*Emiliania*) and 95 % of TC (*Thalassiosira*)."

**Reviewer's comment:** Line 122 & 125: TDN: is this indeed total dissolved nitrogen. But was the inorganic nitrogen not removed by washing three times, so that TDN is more or less DON?

**Reply:** The algae were only washed after thawing – not upon harvesting. Hence, inorganic nitrogen could have been on the cell surfaces at the binning of the washings steps. Since freezing and thawing can induce cell leakage, organic nitrogen from the inner part of the cells could have been released. We agree that this was not sufficiently explained and now changed to:

"These washing steps most likely also entailed a loss of DOM including dissolved organic nitrogen (TON). DOC and TDN (i.e., DIN from remains of culturing medium and DON from cell leakage upon thawing) were measured in the supernatant from the 3 washes…"

**Reviewer's comment:** Line 186: recovered or added labelled phytodetritus?

**Reply:** We added the necessary specifications:

"As will be discussed further, this missing representation of the subsurface sediments in the 4 d *Emiliania* chamber results in an underestimation of the labelled phytodetritus with < 10 % recovered as total processed carbon and uncharacterized OM"

**Reviewer's comment:** Line 267: Freeze-dried algae were measured for total 13C and 15N, thus for coccolith inorganic and organic carbon were combined Both types of carbon will very likely be similarly enriched given the identical carbon sources. What is unclear though is how much of the 13C-DIC attributed to respiration (and recovered in overlying water and pore-water) is from dissolution of the carbonate. Combining 15N-DIN and 13C-DIC release might be give some hints whether 13C from carbonate dissolution matters or not.

**Reply:** This is indeed correct. Carbonate dissolution can contribute to the $^{13}$C-DIC. We therefore added the following paragraph:

"It seems that *Emiliania* OM was initially (4 d and start of 14 d experiment) more respired than *Thalassiosira* (in 4 d experiment: 4 % of the added *Emiliania* OM, of which 3.6 % by DIC release, as opposed to 2 % of the added *Thalassiosira* OM), but this could as well be ascribed to dissolution of the inorganic coccoliths. There was no observable $NH_4^+$ or $NO_x^-$ release as should co-occur with OM mineralization, which would agree with a significant contribution of coccolithophorid dissolution to the observed DIC release."

We also integrated this insight in the paragraph on C:N in the discussion:

Line 619-622: In contrast, the preferred use of nitrogen in the Emiliania 4 d experiment might be masked by the dissolution of the carbonates from the coccoliths, leading to a higher C:N ratio in overlying and pore water as compared to the C:N in the biomass.

**Reviewer's comment:** Line 290-296: I do not see the use of these equations. You present all your data in excess 13C atom fractions. Why then are equation 4, 5, etc needed (these are copypasted from prior work in which del values were reported). Line 289-296 can be deleted without loss of information.

**Reply:** Equations 4-6 are needed for the calculations of bacterial enrichment, since the GC-IRMS results of the FAMES are classically given in delta values, not in atom fractions.

**Reviewer's comment:** All through try to avoid using on the other hand if there is no on the one hand (I have counted it 3 or 4 times).

**Reply:** Thanks for noticing. "On the other hand' has been replaced by synonyms throughout the manuscript.

**Reviewer's comment:** Line 364:.. after which the increase levelled off...... The increase of 13C-DIC...was higher and steady.

**Reply:** We agree that the wording could be improved. The sentence was corrected to:

"In the first three days of both *Emiliania* incubations, $^{13}$C-DIC concentrations quickly accumulated in the overlying water, after which the increase levelled off (Fig. 2 A). The increase of $^{13}$C-DIC in the *Thalassiosira* chambers was higher and steady throughout the 14 d incubation."

**Reviewer's comment:** Line 433: was respired (and recovered in overlying water and pore water).....

**Reply:** We agree and added "(and recovered in overlying water and pore water)" to clarify 'respiration'.

**Reviewer's comment:** Section 4.1: Cold, deep-sea systems sometimes show a delayed response, i.e. low activities during the first two-three days (e.g. Andersson et al 2008 in Arabian Sea). This is one of the reason why your experimental design (two incubation durations) makes sense. Culturing phytoplankton in the lab followed by freeze-drying before additions might perhaps have resulted in the addition of DOC to your experiment at the seafloor. This really depends on the very details of your phytodetritus preparation. Differences in response among studies in the literature can be partly explained by this. Resolution requires 13C measurement of the DOC pool and that is a daunting task.

**Reply:** We included a reference to the work of Andersson et al. (2008) in the Arabian Sea and included the possible effect of a potentially combined POC-DOC addition (lines 499-506) :

"... A delayed response with low activities for a few days was expected as this was also observed in other cold deep-sea ecosystems (e.g., Andersson et al., 2008 in the Arabian Sea) and resulted in the design of the experiment with a shorter and a longer incubation. In case the thawing of the cells resulted in continuous leaking after addition, the relatively slow response may be in part also be explained by the reduced availability of labelled DOM dispersed in the overlying water as opposed to POC at the sediment surface. The share of organic matter provided as DOM and its utilization would have required measurements of the $^{13}$C-labelled DOC pool in the water samples and pore-waters and could not be carried out as part of this study."

**Reviewer's comment:** Line 525: unclear, too cryptic, I understand but will all readers?

**Reply: We further explained this sentence:**

"As the sediment in our study area is well oxygenated in the upper centimeters, settling OM is most likely aerobically mineralized (Donis et al., 2016). Therefore, the probability that obligate anaerobic processes like denitrification take place, is very low. Hence we assume that denitrification of nitrate does not occur in the oxidized sediment layer of our experiments and that the observed accumulation of nitrate in the overlying water is caused solely by nitrification."

**Reviewer's comment:** Line 536: 0.4-1.5% of total nitrification is similar to 0.3-0.4 contribution to sediment ON pool.

**Reply:** We thank the reviewer for pointing this out and added the additional insight to this paragraph:

"Altogether, only 0.4-1.5 % of the total nitrification would then be attributable to nitrification of the ammonium released by the algal detritus, which corresponds to the original addition of algal nitrogen of 0.3-0.4 % to the sediment ON pool."

**Reviewer's comment:** Line 570-572: an additional/alternative explanation. OM delivered to the sediment surface is far away from microbes in the subsurface. Animals, through their bioturbation, mix OM down and deliver OM to bacteria that do not move. This is another component of the (inverted) sediment microbial loop.

**Reply**: We agree with the reviewer. We added the following lines to the discussion (lines 591-594):

"Alternatively, sediment reworking (bioturbation) by mobile fauna redistributes fresh organic matter deposited at the surface to deeper sediment layers, where subsurface bacteria can also access it. This redistribution of fresh carbon was indeed observed in the increase in subsurface algal-derived OM after 14 days (Figure 1A) and higher bacterial assimilation in the sediment subsurface (Figure 3)."

**Reviewer's comment:** Line 588: sediment OM

**Reply:** Agreed, we corrected "POC" to "OM".

**Reviewer's comment:** Line 593: the OM leftovers (POC cannot have a C:N ratio).

**Reply:** Agreed, we corrected "POC" to "the OM leftovers"

**Reviewer's comment:** Line 610-618: another reason for non-closure is that 15N in bacteria and 13C/15N in archaea have not been measured (although the latter might contribute just a minor amount).

**Reply:** This is indeed an interesting additional aspect to cover. We added the following paragraphs (lines 644-654):

"(d) bacterial assimilation of phytodetrital N was not quantified. However, assuming a bacterial C:N ratio of 5 (Goldman and Dennett, 2000) and taking into account that growth of Arctic deep sea bacteria is N-limited (Boetius and Lochte, 1996), it can be expected that bacterial N assimilation was up to 5 times lower than carbon assimilation. This would have doubled the processed share of *Emiliania* detrital N after 14 days (from 6 to 12% of the added N) and almost tripled the processed

share of *Thalassiosira* detrital N after 14 days (from 16 to 42% of the added N). (e) Archaea were not considered in our study, but the experimental duration would probably have been too short, as shown for Thaumarcheota in shallow Icelandic shelf sediments (Lengger et al., 2014). Although sequence data suggest that Archaea contribute only 2-5% to the active members of the benthic prokaryotic community at the study site (Rapp 2018), deep-sea Archaea seem to be involved in protein degradation and carbohydrate metabolism (Li et al., 2015) and especially deep-sea Archaea from high latitudes have been shown to be especially sensitive to changes in food supply (Danovaro et al., 2016). "

**Reviewer's comment:** Line 641: delete p. 201

**Reply:** We thank the reviewer for noticing and corrected accordingly.

**Reviewer's comment:** Line 656: We thank…. … We further thank Anja…

**Reply:** Agreed, corrected.

**Reviewer's comment:** References: balanced coverage, but I missed the Boetius et al. 2012 note in Science.

**Reply:** Thanks for the indeed necessary reference. Boetius et al. (2013) has now been referred to in the introduction in line 57:

"The deposition of phytodetritus from surface water primary production is of crucial importance for the deep-sea benthos (Boetius et al., 2013; Graf, 1989) …"

---

## Referee Comment (RC2) · Anonymous Referee #2 · 12 Oct 2018

Reviewer's comments for bg-2018-264

It is a series of comments for the manuscript, entitled "Carbon and nitrogen turnover in the Arctic deep sea: in situ benthic community response to diatom and coccolithophorid phytodetritus" that has appeared on BG Discussion. I am pleased to read this article with a great interest. Because, this article tries to measure states of both carbon and nitrogen turnover at deep-sea floor through in situ feeding experiments. Even though numbers of experimental trials were a few, it gives an important data for benthic ecosystems research.

[Figure]

I would like to make a couple of comments in terms of this worthy experiments.

1) Why did you select both Thalassiosina and Emiliania sp. for food materials ? Chaeto-celos and Gephylocapsa spp. are also common species of primary production both at middle to high latitude seas. Please ask to add some additional explanation why you use Emiliania and Thalassiosina sp. 2) You have gotten subsamples with syringe tubes. You are better to evaluate statistically how subsamples represent sea floor states. Because, phytodetritus deposition is heterogeneous at sea floor. This introduce patchy distribution of environments as discussed by Glud and others 2009. This may be the same in experimental chamber. 3) You described that diatom frustules are easily de-composed by bacteria according to Bidle and Azam (1999) paper. I suppose that diatom frustules compose of the mixture of organic materials and amorphous silicate. Bacteria may be decomposed organic material. Then silicates dissolve in seawater. Seawater silicates may be undersaturate at Arctic. Do you have any silicate concen-tration data at the experimental site ? 4) I understand that bacteria do not play a big role for dissolution of calcific tests. However, calcite concentration at Arctic is under-saturate in the Arctic deep-sea, coccolith may dissolve quickly at the site. Can you discuss about dissolution procedures of calcareous tests in laboratory condition? It is also required to discuss about Calcite Compensation Depth in Arctic. Normally, dis-solution of calcareous tests at sea floor is much faster at polar seas than temperate oceans. 5) P17, lines 576 $\sim$ 584. This paragraph mainly discuss about foraminiferal assimilation at sea floor. You described that Pyrgo may play a big role for assimilation of organic materials at Hausgarten site. In situ experiments at middle latitude show opportunistic species such as Uvigerina sp. Fursenkoina fusiformis or Epistominella exigua play more big role for assimilating organic materials at sediment water inter-face (for instance, Nomaki et al., 2005, 2008). These species are all size of meiofauna. Main players may not remain on your sieve. Please evaluate more details about roles of foraminifera at sediment-water interface. Series of Nomaki's in situ experimental works at Sagami Bay floor should be helpful to discuss about this topic. 6) One of chamber experiments could only get top cm layer. This means that you are difficult to evaluate

roles of infaunal species at sediment-water interface. It may be helpful to discuss how organisms from deep in sediments assimilate organic materials. You may evaluate thin layer chamber results. Please discuss more details about roles of infaunal species for both carbon and nitrogen turnover through your experimental work.

I am very much appreciated the you are able to respond all the comments properly.

―――――――――――――――――――

---

## Author Comment (AC2) · 16 Oct 2018

**Reply to review of bg-2018-264 by Anonymous Referee #2**

**Reviewer's comment:** It is a series of comments for the manuscript, entitled "Carbon and nitrogen turnover in the Arctic deep sea: in situ benthic community response to diatom and coccolithophorid phytodetritus" that has appeared on BG Discussion. I am pleased to read this article with a great interest. Because, this article tries to measure states of both carbon and nitrogen turnover at deep-sea floor through in situ feeding experiments. Even though numbers of experimental trials were a few, it gives an important data for benthic ecosystems research would like to make a couple of comments in terms of this worthy experiments.

**Authors:** We are very grateful for these constructive comments. Below, we answer to each comment in a point-by-point fashion. We underlined the new text that was added to the manuscript. The line numbers refer to the discussion paper (https://www.biogeosciences-discuss.net/bg-2018-264/bg-2018-264.pdf).

**Reviewer's comment:** Why did you select both Thalassiosina and Emiliania sp. for food materials? Chaetocelos and Gephylocapsa spp. are also common species of primary production both at middle to high latitude seas. Please ask to add some additional explanation why you use Emiliania and Thalassiosina sp.

**Reply:** We selected *Thalassiosira* as it was one of the dominant diatom species in Fram Strait (Nöthig et al. 2015; Bauerfeind et al. 2009, Soltwedel et al. 2015). *Emiliania* was chosen as a possible 'future' food source, since it is a temperate species that can be transported into the Arctic Ocean along with warmer Atlantic currents (Bauerfeind et al., 2009). *Chaetoceros* is indeed another representative for Fram Strait (Nöthig et al. 2015; Bauerfeind et al. 2009, Soltwedel et al. 2015), but its contribution is limited in time and consists mainly of resting spores, not intact cells (Bauerfeind et al. 2009, Lalande et al. 2011). There are no records of dominance of *Gephyrocapsa* sp. The other occurring coccolithophore is *Cocclithus pelagicus,* but this species only occasionally contributed considerably to or dominated the sedimentation of coccolithophorids (Bauerfeind et al. 2009). This is stated in the introduction, lines 44-50, where we specified that *Thalassiosira* spp. dominates the diatom blooms and that *Emiliania huxleyi* has been observed during periods of enhanced Atlantic influence:

"This phenomenon also occurred in the eastern Fram Strait, where previously, phytoplankton communities were typically dominated by diatoms, mainly *Thalassiosira* spp. (Bauerfeind et al., 2009; Lalande et al., 2011). However, during recent warmer years, phytoplankton blooms became more mixed with *Phaeocystis pouchetti* (Nöthig et al., 2015; Soltwedel et al., 2015). Also *Emiliania huxleyi* (Prymnesiophyceae)-dominated coccolithophorid blooms have been observed between 2000-2005 – especially in 2004 – which has been attributed to northward transport of the species into Fram Strait by means of the North Atlantic Current and WSC. This 'Atlantification' with a combined change in water temperature and water mass origin has been suggested as one possible scenario for a community shift in phytoplankton communities in Fram Strait (Bauerfeind et al., 2009)."

**Reviewer's comment:** You have gotten subsamples with syringe tubes. You are better to evaluate statistically how subsamples represent sea floor states. Because, phytodetritus deposition is

heterogeneous at sea floor. This introduce patchy distribution of environments as discussed by Glud and others 2009. This may be the same in experimental chamber.

**Reply:** We actually sampled the entire sediment column recovered by the chamber layer-wise and homogenized the sediment before we took the subsamples. This means that we rule out the possible heterogeneity of the phytodetritus deposition. However, this comment brought our attention to an inconsistency in the discussion (lines 626-628) where we suggested inhomogeneous distribution as a possible explanation why C and N budgets are not closed: "(3) The added phytodetritus might not have been distributed evenly across each benthic chamber. This means that the location within the chamber from which samples for bacterial assimilation and tracer in the pore water were taken will have affected how much C and N was found in faunal and bacterial biomass.". As we assume successful homogenization before sampling this is not a likely reason. Hence, we deleted this sentence from the manuscript.

**Reviewer's comment:** You described that diatom frustules are easily decomposed by bacteria according to Bidle and Azam (1999) paper. I suppose that diatom frustules compose of the mixture of organic materials and amorphous silicate. Bacteria may be decomposed organic material. Then silicates dissolve in seawater. Seawater silicates may be undersaturate at Arctic. Do you have any silicate concentration data at the experimental site?

**Reply:** Thank you for this interesting comment. Seawater silicate is indeed undersaturated in deep Fram Strait waters (~10 µM actual bottom water concentration vs. saturation concentration of silicate ~1000 µM at 2500m water depth and 1.4°C; Sarmiento & Gruber 2013). Hence, silicate dissolution is indeed a pathway to be considered. We added this to the discussion, starting line 552: "Without this organic protection layer, the diatom frustrule rapidly dissolves in undersaturated seawater (Ragueneau et al., 2006) ([$SiO_2$] at our study site ~10 µM vs. $SiO_2$ solubility ~ 1000 µM at 2500 m water depth and 1.4°C (Sarmiento and Gruber, 2013))."

**Reviewer's comment:** I understand that bacteria do not play a big role for dissolution of calcific tests. However, calcite concentration at Arctic is undersaturate in the Arctic deep-sea, coccolith may dissolve quickly at the site. Can you discuss about dissolution procedures of calcareous tests in laboratory condition? It is also required to discuss about Calcite Compensation Depth in Arctic. Normally, dissolution of calcareous tests at sea floor is much faster at polar seas than temperate oceans.

**Reply:** Thank you for pointing this out. We already discuss the dissolution procedures of calcareous tests in lines 552-555, but we now indicate that these results originate from laboratory experiments: "Similarly, the calcite matrix of  the coccoliths can act as a physical barrier against bacterial degradation in laboratory experiments (Engel et al., 2009). However, comparable carbon-specific respiration rates were measured for aggregates of *Emiliania* and *Skeletonema* diatoms, suggesting similar degradability in laboratory experiments using surface waters (Iversen and Ploug, 2010).".

To discuss the results in relation to the position of the lysocline, we added the following paragraph to the discussion (line 514 ff.): "It seems that *Emiliania* OM was initially (4 d and start of 14 d experiment) more respired than *Thalassiosira* (in 4 d experiment: 4 % of the added *Emiliania* OM, of

which 3.6 % by DIC release, as opposed to 2 % of the added *Thalassiosira* OM), but this could as well be ascribed to dissolution of the inorganic coccoliths. There was no $NH_4^+$ or $NO_x^-$ release observed as it should co-occur with OM mineralization. This agree with a significant contribution of coccolithophorid dissolution to the observed DIC release. As the lysocline in the Arctic Ocean lies at ~4000m water depth (Jutterström and Anderson, 2005), if appears unlikely for the calcite from the coccoliths to quickly dissolve at our study site at 2500 m water depth. Nevertheless, Godoi et al. (2009) showed that the release of $CO_2$ during bacterial respiration can cause the decrease of the saturation state of sea water in the cell's microenvironment and may hence favour $CaCO_3$ dissolution."

**Reviewer's comment:** P17, lines 576~584. This paragraph mainly discuss about foraminiferal assimilation at sea floor. You described that Pyrgo may play a big role for assimilation of organic materials at Hausgarten site. In situ experiments at middle latitude show opportunistic species such as Uvigerina sp. Fursenkoina fusiformis or Epistominella exigua play more big role for assimilating organic materials at sediment water interface (for instance, Nomaki et al., 2005, 2008). These species are all size of meiofauna. Main players may not remain on your sieve. Please evaluate more details about roles of foraminifera at sediment-water interface. Series of Nomaki's in situ experimental works at Sagami Bay floor should be helpful to discuss about this topic.

**Reply:** The meiofaunal foraminifera contribution to carbon and nitrogen cycling is indeed an interesting topic to discuss, since we only considered the larger (>250 μm) foraminifera in our study. We highlight this now in the discussion, line 572-580: "Foraminifera still had a two orders of magnitude higher carbon-specific assimilation than bacteria, implying that larger organisms continued to dominate the competition for fresh OM. This confirms earlier studies showing that foraminifera can be key players in the early diagenesis of fresh OM at the deep-sea floor (Moodley et al., 2000, 2002; Nomaki et al., 2005; Woulds et al., 2007). However, these studies also included the meiofauna fraction of foraminifera (63-250μm). Although macrofaunal (>250μm) foraminifera can have a retarded response to phytodetritus inputs as compared to smaller (> 63 μm) foraminifera (Sweetman et al., 2009), the carbon assimilation rate by macrofaunal foraminifera in this study is similar to that of smaller foraminifera at Station M (Enge et al., 2011) and the central basin of Sagami bay (1449m) (Nomaki et al., 2005). Nevertheless, as smaller foraminifera were not analyzed here, it may be that the overall assimilation of this group was still underestimated."

We also highlight the importance of meiofaunal foraminifera in the discussion of the mass budget (line 610 ff.): "(c) We did not consider meiobenthos < 250 μm, since nematodes, the most abundant metazoan component of deep-sea meiobenthos, are usually responsible for only < 1 % of the total mineralization (Ingels et al., 2010). However, the meiofauna fraction of the foraminifera could have contributed to the mineralization (Moodley et al., 2002; Nomaki et al., 2005)"

**Reviewer's comment:** One of chamber experiments could only get top cm layer. This means that you are difficult to evaluate roles of infaunal species at sediment-water interface. It may be helpful to discuss how organisms from deep in sediments assimilate organic materials. You may evaluate thin layer chamber results. Please discuss more details about roles of infaunal species for both carbon and nitrogen turnover through your experimental work.

**Reply:** Indeed, we discuss this in line 608-610: "(a) We could not sample the sediment subsurface layers of the 4 d Emiliania experiment as deeper layers were lost upon retrieval of samples from the chambers on board, and as such miss the subsurface processing pathway. However, in the other experiments, this part accounts for < 10 % of the sum of the total processed carbon and uncharacterized OM."

This means that most (>90%) of the carbon and nitrogen is turned over at the sediment surface (thin layer) and that the role for deeper infauna is limited (<<10%, because this 10% mainly includes the uncharacterized OM). However, biogenic mixing by infauna (bioturbation) is the process that must have brought the fresh detritus deeper down.

In accordance with the comments of the first reviewer, Jack Middelburg, we added the following paragraph that we believe also addresses the comments of Anonymous Reviewer #2 (line 572 ff.):

"Alternatively, sediment reworking (bioturbation) by infauna also redistributes fresh organic matter deposited at the surface to the deeper sediment layers, where subsurface bacteria can also access it. Particle mixing in Arctic sediments is usually limited to the upper 3 cm of the sediment (Clough et al. 1997, Morata et al. 2015, Krauss 2016), which fits with our observations on the increase in subsurface algal-derived OM after 14 days (Figure 1A) and higher bacterial assimilation in the sediment subsurface (Figure 3)."